# Sum-Product-Set Networks: Deep Tractable Models for Tree-Structured Graphs

**Milan Papež, Martin Rektoris, Václav Šmídl & Tomáš Pevný**
Artificial Intelligence Center, Czech Technical University
{papezmil,rektomar,smidlva1,pevnytom}@fel.cvut.cz

## Abstract

Daily internet communication relies heavily on tree-structured graphs, embodied by popular data formats such as XML and JSON. However, many recent generative (probabilistic) models utilize neural networks to learn a probability distribution over undirected cyclic graphs. This assumption of a generic graph structure brings various computational challenges, and, more importantly, the presence of non-linearities in neural networks does not permit tractable probabilistic inference. We address these problems by proposing sum-product-set networks, an extension of probabilistic circuits from unstructured tensor data to tree-structured graph data. To this end, we use random finite sets to reflect a variable number of nodes and edges in the graph and to allow for exact and efficient inference. We demonstrate that our tractable model performs comparably to various intractable models based on neural networks.

## 1 Introduction

One of the essential paradigm shifts in artificial intelligence and machine learning over the last years has been the transition from probabilistic models over fixed-size unstructured data (tensors) to probabilistic models over variable-size structured data (graphs) (Bronstein et al., 2017; Wu et al., 2021). Tree-structured data are a specific type of generic graph-structured data that describe real or abstract objects (vertices) and their hierarchical relations (edges). These data structures appear in many scientific domains, including cheminformatics (Bianucci et al., 2000), physics (Kahn et al., 2022), and natural language processing (Ma et al., 2018). They are used by humans in data-collecting mechanisms to organize knowledge into various machine-generated formats, such as JSON (Pezoa et al., 2016), XML (Tekli et al., 2016) and YAML (Ben-Kiki et al., 2009) to mention a few. Designing a probabilistic model for these tree-structured file formats is one of the key motivations of this paper.

The development of models for tree-structured data has been thoroughly, but almost exclusively, pursued in the NLP domain (Tai et al., 2015; Zhou et al., 2016; Cheng et al., 2018; Ma et al., 2018). Unfortunately, these models rely solely on variants of neural networks (NNs), are non-generative, and lack clear probabilistic interpretation. There has also recently been growing interest in designing generative models for general graph-structured data (Simonovsky & Komodakis, 2018; De Cao & Kipf, 2018; You et al., 2018; Jo et al., 2022; Luo et al., 2021). However, the underlying principle of these generic models is to perform the message passing over a neighborhood of each node in the graph, meaning that they visit many of the nodes several times. Directly applying them to the trees and not adapting them to respect the parent-child ancestry of the trees would incur unnecessary computational costs. Additionally, these models assume that the features are assigned to each node and all have the same dimension. More importantly, they preclude tractable probabilistic inference, necessitating approximate techniques to answer even the most basic queries.

In sensitive applications (e.g., healthcare, finance, and cybersecurity), there is an increasing legal concern about providing non-approximate and fast decision-making. Probabilistic circuits (PCs) (Vergari et al., 2020) are *tractable* probabilistic (generative) models that guarantee to answer a large family of complex probabilistic queries (Vergari et al., 2021) exactly and efficiently. For instance, the marginal queries form the fundamental part of many more advanced queries. Their practicality lies in allowing us to consistently address various tasks, including dealing with missing data (Peharz et al., 2020) and explaining anomalous samples (Lüdtke et al., 2023). This paper is interested in

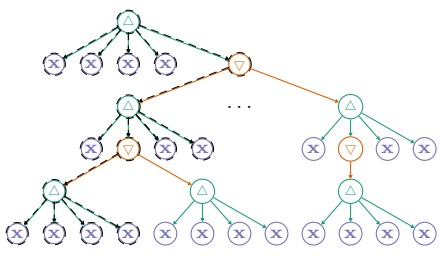

```
{
  "ind1": 1, "lumo": -1.246, "inda": 0, "logp": 4.23, "atoms": [
    {
      "element": "c",
      "bonds": [
        {"element": "c", "charge": -0.117, "bond": 7, "atom": 22},
        {"element": "h", "charge":  0.142, "bond": 1, "atom":  3}
      ],
      "charge": -0.117,
      "atom": 22
    },
        :
        :
    {
      "element": "h",
      "bonds": [
        {"element": "c", "charge": -0.117, "bond": 1, "atom": 22}
      ],
      "charge": 0.142,
      "atom": 3
    }
  ]
}
```

Figure 1: *Tree-structured graphs.* Left: an example of a molecule from the Mutagenesis dataset (Debnath et al., 1991) encoded in the JSON format. Right: the corresponding, tree-structured graph, $T$ (Definition 1), describing relations between the atoms and their properties, and its *schema* (dashed line), $S$ (Definition 2). Here, $\triangle$ is the heterogeneous node (green), $\triangledown$ is the homogeneous node (orange), and $\mathbf{x}$ is the leaf node (blue).

a specific sub-type of PCs—sum-product networks (SPNs)—which represent a probability density over fixed-size unstructured data (Poon & Domingos, 2011).

To our knowledge, no SPN is designed to represent a probability distribution over a variable-size, tree-structured graph. Here, it is essential to note that an SPN is also a graph. To distinguish between the two graphs, we refer to the former as the *data* graph and the latter as the *computational* graph. Similarly, to distinguish between the vertices of these two graphs, we refer to vertices of the data graph and computational graph as data *nodes* and computational *units*, respectively. We propose a new SPN (i.e., a new type of PCs) by seeing the data graph as a recursive hierarchy of sets, where each parent data node is a set of its child data nodes. Unlike the aforementioned models for the undirected cyclic graphs, our model respects the parent-child direction when processing the data graph. We use the theory of random finite sets (Nguyen, 2006) to induce a probability distribution over repeating data subgraphs, i.e., sets with identical properties. This allows us to extend the computational graph with an original computational unit, a *set* unit. Our model provides an efficient sampling of new data graphs and exact marginal inference over selected data nodes, which allows us to deal with missing parts of the JSON files. It also permits the data nodes to have heterogeneous features, where each node can represent data of different dimensions and modalities.

In summary, this paper offers the following contributions:

- We propose sum-product-set networks (SPSNs), extending the predominant focus of PCs from unstructured tensor data to tree-structured graph data (Section 3).
- We show that SPSNs respecting the standard structural constraints of PCs are tractable under mild assumptions on the set unit (Section 3.1).
- We investigate the exchangeability of SPSNs, concluding that SPSNs are permutation invariant under the reordering of the arguments in the set units and input units (if the input units admit some form of exchangeability) and that the invariance propagates through SPSNs fundamentally based on the structural constraints (Section 3.2).
- We show that SPSNs deliver competitive performance to intractable models relying on much more densely connected and highly nonlinear NNs, which are unequipped to provide exact answers to probabilistic queries (Section 5).

## 2 TREE-STRUCTURED DATA

A single instance of tree-structured, heterogeneous data is given by an attributed data graph, $T$. In contrast to a fixed-size, *unstructured*, random variable, $\mathbf{x} = (x_1, \ldots, x_n) \in \mathcal{X} \subseteq \mathbb{R}^n$, this graph forms a hierarchy of random-size sets.

**Definition 1.** *(Data graph).* $T := (V, E, X)$ *is an attributed, tree-structured graph, where $V$ is a set of vertices, $E$ is a set of edges, and $X$ is a set of attributes (features). $V := (L, H, O)$ splits into three subsets of data nodes: leaf nodes, $L$, heterogeneous nodes, $H$, and homogeneous nodes, $O$. Let $\mathbf{ch}(v)$ and $\mathbf{pa}(v)$ denote the set of child and parent nodes of $v \in V$, respectively. All elements of $\mathbf{ch}(v)$ are of an identical type if $v \in O$, and some or all elements of $\mathbf{ch}(v)$ are of a different type*

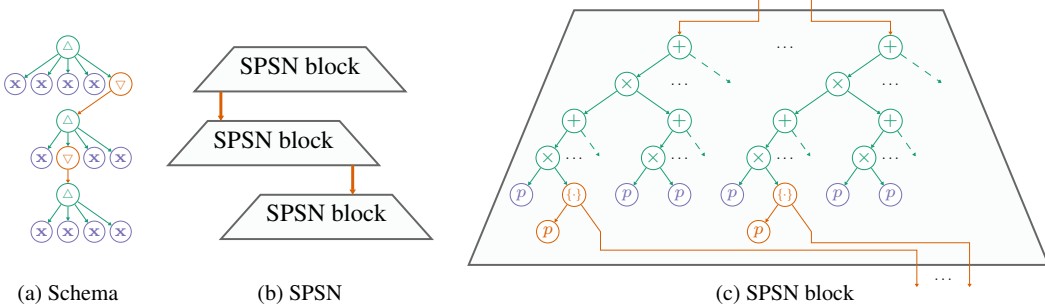

Figure 2: *Sum-product-set networks.* (a) The schema (Definition 2) from the example in Figure 1. (b) The SPSN network comprises SPSN blocks that correspondingly model the heterogeneous nodes depicted in (a). (c) The SPSN block embodies the computational (sub)graph, $\mathcal{G}$, (Definition 3), where $+$, $\times$, $\{\cdot\}$ and $p$ are the sum unit, product unit, *set* unit and input unit of $\mathcal{G}$, respectively. The elements of the heterogeneous node in (a) are modeled by the corresponding parts of $\mathcal{G}$ in (c), as displayed in green, orange, and blue.

*if $v \in H$. We assume that only the leaf nodes are attributed by $\mathbf{x}_v \in \mathcal{X}_v \subseteq \mathbb{R}^{n_v}$, with possibly different space and its dimension for each $v \in L$.*

**Definition 2.** *(Schema). Let $T$ be a tree-structured, heterogeneous data graph (Definition 1). Then, a subtree, $S$, which results from $T$ by following all children of each $v \in H$ and only one child of each $v \in O$—such that it allows us to reach the deepest level of $T$—is referred to as the* schema[1].

We model each heterogeneous node, $T_v := (T_{w_1}, \ldots, T_{w_m}) \in \mathcal{T}_v$, $v \in H$, as a finite set. It is an ordered set of features or other sets, $T_w \in \mathcal{T}_w$, where the elements are random but their number, $m \in \mathbb{N}_0$, is fixed. For $v \in H$, $\mathcal{T}_v$ is the Cartesian product space composed of the spaces corresponding to the elements of $T_v$. Importantly, we propose to model each homogeneous node, $T_v := \{T_{w_1}, \ldots, T_{w_m}\} \in \mathcal{T}_v$, $v \in O$, as a finite random set (RFS), i.e., a simple, finite point process (Van Lieshout, 2000; Daley et al., 2003; Nguyen, 2006; Mahler, 2007). This is an unordered set of *distinct* features or other sets, such that not only the individual elements, $T_w \in \mathcal{T}_w$, are random, but also their number, $m \in \mathbb{N}_0$, is random. For $v \in O$, $\mathcal{T}_v := \mathcal{F}(\mathcal{T}_w)$ is the hyperspace of all finite subsets of some underlying space $\mathcal{T}_w$ (which is common to all elements). We refer to Section B for more details on RFSs. The leaf node, $T_v := \mathbf{x}_v = (x_1 \ldots, x_m) \in \mathcal{T}_v$, $v \in L$, contains a feature vector (i.e., it is also a finite set). For $v \in L$, $\mathcal{T}_v$ is usually a subspace of $\mathbb{R}^{m_v}$. We show an example of a single instance of tree-structured data in Figure 1, where $T = (x, x, x, x, \{(x, \{(x, x, x, x), (x, x, x, x)\}, x, x), \ldots, (x, \{(x, x, x, x)\}, x, x)\})$. The whole tree $T$ lives in a hybrid space $\mathcal{T}$, i.e., a product space composed of continuous spaces, discrete spaces, and hyperspaces of other RFSs. Each $T$ is thus a hierarchy of several RFSs. Indeed, such constructions are possible and are used to define "random cluster processes" (Mahler, 2001; Mahler & MN, 2002).

It follows from Definition 1 and Definition 2 that, for heterogeneous nodes, $v \in H$, each child subtree, $T_w$, has a different schema for all $w \in \mathbf{ch}(v)$; whereas for homogeneous nodes, $v \in O$, each child, $T_w$, has the same schema for all $w \in \mathbf{ch}(v)$. This further implies that $T$ can be defined recursively by a subtree, $T_v := [T_{w_1} \ldots, T_{w_m}]$, rooted at $v \in V$, where the parentheses $[\cdot]$ instantiate into $\{\cdot\}$ for homogeneous nodes, and into $(\cdot)$ for heterogeneous nodes, to distinguish if the number of elements is random or not. Note also that the number of elements in homogeneous nodes, $v \in O$, differs for each instance of $T$, while it remains the same for all heterogeneous nodes, $v \in H$.

**Problem definition.** Our objective is to learn a probability density over tree-structured graphs (Definition 1), $p(T)$, given a collection of observed graphs $\{T_1, \ldots, T_m\}$, where each $T_i$ contains a different number of vertices and edges but follows the same schema.

---

[1]Recall that one the key motivations of this paper is to design a probabilistic model for the tree-structured file formats (Section 1). The intuition behind Definition 2 follows from the schema of the JSON files (Pezoa et al., 2016). We do not consider the schema used in the relational databases.

## 3 SUM-PRODUCT-SET NETWORKS

A sum-product-set network (SPSN) is a probability density over a tree-structured graph, $p(T)$. This differs from the conventional SPN (Poon & Domingos, 2011), which is a probability density over the unstructured vector data, $p(\mathbf{x})$. For a short introduction to SPNs, see Section A in the supplementary material. We define an SPSN by a parameterized computational graph, $\mathcal{G}$, and a scope function, $\psi$.

**Definition 3.** *(Computational graph). $\mathcal{G} \coloneqq (\mathcal{V}, \mathcal{E}, \theta)$ is a parameterized, directed, acyclic graph, where $\mathcal{V}$ is a set of vertices, $\mathcal{E}$ is set of edges, and $\theta \in \Theta$ are parameters. $\mathcal{V} \coloneqq (\mathsf{S}, \mathsf{P}, \mathsf{B}, \mathsf{I})$ contains four subsets of computational units: sum units, $\mathsf{S}$, product units, $\mathsf{P}$, set units, $\mathsf{B}$, and input units, $\mathsf{I}$. The sum units and product units have multiple children; however, as detailed later, the set unit has only two children, $\mathbf{ch}(u) \coloneqq (a, b)$, $u \in \mathsf{B}$. $\theta$ contains parameters of sum units, i.e., non-negative and normalized weights, $(w_{u,c})_{c \in \mathbf{ch}(u)}$, $w_{u,c} \geq 0$, $\sum_{c \in \mathbf{ch}(u)} w_{u,c} = 1$, $u \in \mathsf{S}$, and parameters of input units which are specific to possibly different densities.*

**Definition 4.** *(Scope function). The mapping $\psi_u : \mathcal{V} \to \mathcal{F}(T)$—from the set of units to the power set of $T \in \mathcal{T}$—outputs a subset of $T$ for each $u \in \mathcal{V}$ and is referred to as the* scope function*. If $u$ is the root unit, then $\psi_u = T$. If $u$ is a sum unit, product unit, or set unit, then $\psi_u = \bigcup_{c \in \mathbf{ch}(u)} \psi_c$.*

Each unit of the computational graph (Definition 3), $u \in \mathcal{V}$, induces a probability density over a given (subset of) node(s) of the data graph (Definition 1), $v \in V$. The functionality of this density, $p_u(T_v)$, depends on the type of the computational unit.

The **sum unit** computes the mixture density, $p_u(T_v) = \sum_{c \in \mathbf{ch}(u)} w_{u,c} p_c(T_v)$, $u \in \mathsf{S}$, $v \in (L, H)$, where $w_{u,c}$ is the weight connecting the sum unit with a child unit. The **product unit** computes the factored density, $p_u(T_v) = \prod_{c \in \mathbf{ch}(u)} p_c(\psi_c)$, $u \in \mathsf{P}$, $v \in (L, H)$. It introduces conditional independence among the scopes of its children, $\psi_c$, establishing unweighted connections between this unit and its child units. The **input unit** computes a user-defined probability density, $p_u(x_v)$, $u \in \mathsf{I}$, $v \in L$. It is defined over a subset $x_v$ of $T_v \coloneqq \mathbf{x}_v$, corresponding to the scope, $\psi_u$, which can be univariate or multivariate (Peharz et al., 2015).

The newly introduced **set unit** computes a probability density of an RFS,

$$p_u(T_v) = p(m) c^m m! p(T_{w_1}, \ldots, T_{w_m}), \tag{1}$$

$u \in \mathsf{B}$, $v \in O$, where $p(m)$ is the cardinality distribution, and $p(T_{w_1}, \ldots, T_{w_m})$ is the feature density (conditioned on $m$). These are the two children of the set unit (Definition 3), spanning computational subgraphs on their own (Figure 2). The proportionality factor, $m! = \prod_{i=1}^{m} i$, comes from the symmetry of $p(T_{w_1}, \ldots, T_{w_m})$. It reflects the fact that $p(T_{w_1}, \ldots, T_{w_m})$ is permutation invariant, i.e., it gives the same value to all $m!$ possible permutations of $\{T_{w_1}, \ldots, T_{w_m}\}$. Recall from Section 2 that $T_v \in \mathcal{F}(\mathcal{T}_w)$, where $\mathcal{T}_w$ is the underlying space of the RFS. $c$ is a constant with units of hyper-volume of $\mathcal{T}_w$, which ensures that (1) is unit-less by canceling out the units of $p(T_{w_1}, \ldots, T_{w_m})$ with the units of $c^m$. This mechanism is important when computing integrals with (1). Section B provides more details on the subject of integrating functions of RFSs. Note that, contrary to the other units, the set unit is defined only for the homogeneous node, $v \in O$.

**Assumption 1.** *(Requirements on the set unit). We make the following requirements on the probability density of the set unit: (a) each element of $T_v \coloneqq \{T_{w_1}, \ldots, T_{w_m}\}$ resides in the same space, i.e., $T_w \in \mathcal{T}_w$, for all $w \in \mathbf{ch}(v)$; (b) the realizations $\{T_{w_1}, \ldots, T_{w_m}\}$ are distinct; (c) the cardinality distribution $p(m) = 0$ for a sufficiently large $m$; and (d) the elements $\{T_{w_1}, \ldots, T_{w_m}\}$ are independent and identically distributed (i.i.d.).*

Assumptions 1(a-c) are the standard assumptions on RFSs (Section B). Assumption 1(d) reduces the feature density in (1) to the product of $m$ densities over the identical scope, $p(T_{w_1}, \ldots, T_{w_m}) = \prod_{i=i}^{m} p(T_{w_i})$ (i.e., not the disjoint scope and, therefore, it is not the product unit), where each $p(T_{w_i})$ is indexed by the same set of parameters. Consequently, the feature density treats $\{T_{w_i}\}_{i=i}^{m}$ as i.i.d. instances, aggregating them by the product of densities. Note that, if Assumption 1(d) holds, and $p(m)$ is the Poisson distribution, then (1) is the Poisson point process (Grimmett & Stirzaker, 2001). We provide more comments on Assumption 1(d), including a possible way to relax it, in Remark 2.

**Constructing SPSNs.** We illustrate the design of SPSNs in Figure 2. The construction begins with extracting the schema of the data graph (Figure 2(a) which corresponds to the example of $T$ in Figure 1). Then, starting from the top of the schema, we create an SPSN block for each heterogeneous

node (Figure 2(b)). The SPSN block (Figure 2(c)) alters many layers of sum units and product units (green). Every time there is a product layer, the heterogeneous node, $T_v := (T_{w_1}, \ldots, T_{w_m})$, is split into two (or multiple depending on the number of children of the product units) parts, $T_{v_1}$ and $T_{v_2}$. This process is repeated recursively until $T_{v_1}$ and $T_{v_2}$ are either singletons or subsets that result from a user-defined limit on the maximum number of product layers in the block. Each of these subsets or singletons is then modeled by the input unit (blue). Importantly, this reduction always has to separate all homogeneous nodes out of the heterogeneous node, $T_v$, as single elements modeled by the *set* unit (orange). Note that the sum units create duplicate parts of the computational graph (children have an identical scope), which we leave out in our illustration for simplicity (the dashed line). The consequence is that there will be multiple set units. We gather all edges leading from the feature density of these set units and connect them to the block (Figure 2(b)) modeling the subsequent heterogeneous node in the schema (Figure 2(a)). We provide a detailed algorithm to construct SPSNs in Section E, along with a simple example without the block structures (Figure 4).

**Hyper-parameters.** The key hyper-parameter is the number of layers of the SPSN block, $n_l$. We consider that a single SPSN layer comprises one sum layer and one product layer. The other hyper-parameters are the number of children of all sum units, $n_s$, and product units, $n_p$, which are common across all layers ($n_l = 2$, $n_s = 2$, and $n_p = 2$ in Figure 2(c)).

## 3.1 TRACTABILITY

Tractability is the cornerstone of PCs. A PC is tractable if it performs probabilistic inference *exactly* and *efficiently* (Choi et al., 2020; Vergari et al., 2021). In other words, the probabilistic queries are answered without approximation (or heuristics) and in time, which is polynomial in the number of edges of the computational graph. Various standard probabilistic queries (e.g., moments, marginal queries, and conditional queries) can be defined in terms of the following integral:

$$\nu(f) = \int f(T) p(T) \nu(dT), \tag{2}$$

where $f : \mathcal{T} \to \mathbb{R}$ is a function that allow us to formulate probabilistic queries, and $\nu$ is a reference measure on $\mathcal{T}$ (Section 2). The composite nature of $\mathcal{T}$ imposes a rather complex structure on $\nu$. It contains measures specifically tailored for RFSs, which makes the integration different compared to the standard Lebesgue measure (see Section B and Section D.2 for details).

**Structural constraints.** If (2) admits an algebraically closed-form solution, then SPSNs are tractable. This is what we demonstrate in this section. To this end, both $p$ and $f$ have to satisfy certain restrictions on their structure, which we present in Definition 5 and Definition 6.

**Definition 5.** *(Structural constraints on $p$). We consider the following constraints on $\mathcal{G}$.* **Smoothness:** *children of any sum unit have the same scopes, i.e., each $u \in \mathsf{S}$ satisfies $\forall a, b \in \mathbf{ch}(u) : \psi_a = \psi_b$.* **Decomposability:** *children of any product unit have pairwise disjoint scopes, i.e., each $u \in \mathsf{P}$ satisfies $\forall a, b \in \mathbf{ch}(u) : \psi_a \cap \psi_b = \varnothing$.*

The SPSNs inherit the standard structural constraints used in PCs (Shen et al., 2016; Vergari et al., 2021). The set unit does not violate these constraints since the cardinality distribution and the feature density are computational subgraphs given by the SPSN units (Definition 3).

**Definition 6.** *(Structural constrains on $f$.) Let $f(T) := \prod_{u \in \mathsf{L}} f_u(\psi_u)$ be a factorization of $f$, where $(\psi_u)_{u \in \mathsf{L}}$ are pairwise disjoint scopes that are unique among all the input units.*

Note that the scopes of all input units, $(\psi_u)_{u \in \mathsf{I}}$, form a multiset since there are usually repeating scopes among all $u \in \mathsf{I}$. These repetitions result from the presence of sum units in the computational graph, as they have children with identical scopes (Definition 5). The collection $(\psi_u)_{u \in \mathsf{L}}$, on the other hand, contains no repeating elements, i.e., $\mathsf{L} \subseteq \mathsf{I}$, are the input units with a unique scope. We obtain this set if we follow only a single child of each sum unit when traversing the computational graph from the root to the inputs (as in the induced tree (Zhao et al., 2016; Trapp et al., 2019)).

Definition 6 allows us to target an arbitrary part of the data graph, $T$, which is spanned from a given data node, $v \in V$, (Definition 1). That is, we can define $T := T_{-v} \cup T_v$, where $T_v$ is composed of the subsets of $T$ that are reachable from $v$, and $T_{-v}$ is the complement. Consider $f(T) := \mathbb{1}_A(T)$, where the indicator function $\mathbb{1}_A(T) := 1$ if $T \in A$, for a measurable subset $A \subseteq \mathcal{T}$, or $\mathbb{1}_A(T) := 0$ otherwise. Now, let $A := E_{-v} \times A_v$, where $E_{-v} \in \mathcal{T}_{-v}$ is an evidence assignment (a specific

realization) corresponding to $T_{-v}$, and $A_v \subseteq \mathcal{T}_v$ is a measurable subset corresponding to $T_v$. Then, the integral (2) yields the *marginal query* $P(E_{-v}, A_v)$. This query is useful if there is a (subset of) node(s) with *missing values*, e.g., a leaf node, $v \in L$, which we demonstrate in Section 5.

**Proposition 1.** *(Tractability of SPSNs). Let $p(T)$ be an SPSN satisfying Assumption 1 and Definition 5, and let $f(T)$ be a function satisfying Definition 6. Then, the integral (2) is tractable and can be computed recursively as follows:*

$$I_u = \begin{cases} \sum_{k=0}^{\infty} p(k) \prod_{i=1}^{k} I_i, & \text{for } u \in \mathsf{B}, \\ \sum_{c \in \mathbf{ch}(u)} w_{u,c} I_c, & \text{for } u \in \mathsf{S}, \\ \prod_{c \in \mathbf{ch}(u)} I_c, & \text{for } u \in \mathsf{P}, \\ \int f_u(\psi_u) p_u(\psi_u) \nu_u(d\psi_u), & \text{for } u \in \mathsf{I}, \end{cases}$$

*where the measure $\nu_u(d\psi_u)$ instantiates itself either as the Lebesgue measure or the counting measure, depending on the specific form of the scope $\psi_u$.*

*Proof.* See Section D.2 in the supplementary material. $\qquad\square$

Proposition 1 starts the integration by finding a closed-form solution for the integrals w.r.t. the input units, $u \in \mathsf{I}$. The results, $I_u$, are then recursively propagated in the feed-forward pass through the computational graph and are simply aggregated based on the rules characteristic to the sum, product, and set unit. Specifically, the integration passes through the set unit similarly to the sum and product units. It computes an algebraically closed-form solution consisting of a weighted sum of products of integrals passed from the feature density (i.e., a tractable sub-SPSN). Note that the integration reduces to the one used in the conventional SPNs if there are no set units, see Proposition 5.

The infinite sum in the aggregation rule of the set unit ($u \in \mathsf{B}$) in Proposition 1 might give an impression that SPSNs are intractable. However, recall from Assumption 1 that $p(k) = 0$ for a sufficiently large $k$ (consider, e.g., the Poisson distribution), and the infinite sum therefore becomes a finite one (Remark 1). This is commonly the case in practice since $p(k)$ is learned from collections of graphs with a finite number of edges.

## 3.2 EXCHANGEBILITY

The study of probabilistic symmetries has attracted increased attention in the neural network literature (Bloem-Reddy & Teh, 2020). On the other hand, the exchangeability of PCs has been investigated marginally. The relational SPNs (Nath & Domingos, 2015) and the exchangeability-aware SPNs (Lüdtke et al., 2022) are, to the best of our knowledge, the only examples of PCs introducing exchangeable computational units. However, none of them answers the fundamental question about exchangeability: Under what constraints is it possible to permute the arguments of a PC?

To define the notion of finite *full* exchangeability of a probability density (see Section C for details), we use the finite symmetric group of a set of $n$ elements, $\mathbb{S}_n$. This is a set of all $n!$ permutations of $[n] \coloneqq (1, \ldots, n)$, and, any of its members, $\pi \in \mathbb{S}_n$, exchanges the elements of an $n$-dimensional vector, $\mathbf{x} \coloneqq (x_1, \ldots, x_n)$, in the following way: $\pi \cdot \mathbf{x} = (x_{\pi(1)}, \ldots, x_{\pi(n)})$.

**Definition 7.** *(Full exchangeability). The probability density $p$ is fully exchangeable iff $p(\mathbf{x}) = p(\pi \cdot \mathbf{x})$ for all $\pi \in \mathbb{S}_n$. We say that $\mathbf{x}$ is fully exchangeable if $p(\mathbf{x})$ is.*

The full exchangeability (complete probabilistic symmetry) is sometimes unrealistic in practical applications. The relaxed notion of finite *partial* exchangeability (de Finetti, 1937; Aldous, 1981; Diaconis & Freedman, 1984; Diaconis, 1988a;b) admits the existence of several different and related groups where full exchangeability applies within each group but not across the groups.

To describe the partial exchangeability, we rely on the product of $m$ finite symmetric groups, $\mathbb{S}_{\mathbf{n}_m} \coloneqq \mathbb{S}_{n_1} \times \cdots \times \mathbb{S}_{n_m}$, where $\mathbf{n}_m \coloneqq (n_1, \ldots, n_m)$. Any member, $\boldsymbol{\pi} \in \mathbb{S}_{\mathbf{n}_m}$, permutes each of $m$ elements in the collection, $X \coloneqq (\mathbf{x}_1, \ldots, \mathbf{x}_m)$, individually as follows: $\boldsymbol{\pi} \cdot X = (\pi_1 \cdot \mathbf{x}_1, \ldots, \pi_m \cdot \mathbf{x}_m)$.

**Definition 8.** *(Partial exchangeability). The probability density $p$ is partially exchangeable iff $p(X) = p(\boldsymbol{\pi} \cdot X)$ for all $\boldsymbol{\pi} \in \mathbb{S}_{\mathbf{n}_m}$. We say that $X$ is partially exchangeable if $p(X)$ is.*

Definition 8 allows us to study the exchangeability of probabilistic models in situations where they follow structural limitations that prevent the direct use of full exchangeability. SPSNs respecting

Assumption 1 and Definition 5 have a constrained computational graph, which imposes limitations on their input-output behavior. Therefore, it does not apply that exchanging any two nodes in the data graph, $T$, has no impact on the value of $p(T)$. We present Proposition 3 to describe under what restrictions the data nodes can be exchanged, how the permutations propagate through the computational graph, and in what sense the exchangeability affects the different types of computational units.

**Proposition 2.** *(Structurally constrained permutations.) Consider a PC satisfying Definition 5. Then, $q_a \in \mathcal{Q}$ is a permutation operator which targets a specific computational unit, $a \in \mathcal{V}$, and propagates through the computational graph $\mathcal{G}$—from the root to the inputs—in the following way:*

$$p_u(q_a \cdot \psi_u) = \begin{cases} \sum_{c \in \mathbf{ch}(u)} w_{u,c} p_c(q_a \cdot \psi_u), & \text{for } a \neq u \text{ and } u \in \mathsf{S}, \\ \prod_{v \in \underline{\mathbf{ch}}_a(u)} p_v(q_v \cdot \psi_v) \prod_{c \in \overline{\mathbf{ch}}_a(u)} p_c(\psi_c), & \text{for } a \neq u \text{ and } u \in \mathsf{P}, \\ p_u(\boldsymbol{\pi} \cdot \psi_u), & \text{for } a = u \text{ and } u \notin \{\mathsf{S}, \mathsf{P}\}, \end{cases}$$

*where $\underline{\mathbf{ch}}_a(u)$ and $\overline{\mathbf{ch}}_a(u)$ are children of $u$ that are and are not the ancestors of $a$, respectively. Consequently, $\mathcal{Q}$ is a group that results from the structural restrictions on the computational graph $\mathcal{G}$.*

*Proof.* See Section C.1. □

Proposition 2 shows that $q_a$ propagates through each sum unit to all its children, passes through each product unit only to those children that are the ancestors of $a$, and instantiates itself to the permutation, $\boldsymbol{\pi}$, when reaching the targeted unit, $a$. This recursive mechanism allows us to formulate the exchangeability of SPSNs in Proposition 3.

**Proposition 3.** *(Exchangeability of SPSNs). Let $p(T)$ be an SPSN satisfying Assumption 1 and Definition 5. Let $\mathsf{I}_\mathsf{E} \in \mathsf{I}$ be a subset of input units that are exchangeable in the sense of Definition 7 or Definition 8. Then, the SPSN is partially exchangeable, $p(q_a \cdot T) = p(T)$, for each $a \in \{\mathsf{I}_\mathsf{E}, \mathsf{B}\}$.*

*Proof.* See Section C.2. □

Proposition 3 states that changing the order of the arguments corresponding to *exchangeable* input units and set units does not influence the resulting value of $p(T)$, i.e., the SPSNs are invariant under the reordering of the elements in the scopes of these units. There must be at least one input unit that is multivariate and exchangeable in the sense of Definition 7 or Definition 8 to satisfy exchangebility w.r.t. $\mathsf{I}_\mathsf{E}$; otherwise, SPSNs are exchangeable only w.r.t. $\mathsf{B}$. This implies that $T$ can always be exchanged w.r.t. the homogeneous nodes and only the leaf nodes admitting exchangeability. The heterogeneous nodes are not exchangeable. Full exchangeability is possible only when there are no product units, and the input units are fully exchangeable. The presence of product units thus always imposes partial exchangeability. The structural constraints of SPSNs impose a specific type of probabilistic symmetry. In other words, an SPSN can be seen as a probabilistic symmetry structure invariant under the action of a group, $\mathcal{Q}$, resulting from the connections in the computational graph.

## 4 RELATED WORK

**Non-probabilistic models (NPMs).** Graph neural networks (GNNs) have become a powerful approach for non-probabilistic representation learning on graphs. Variants of GNNs range from their original formulation (Gori et al., 2005; Scarselli et al., 2008) to GCN (Kipf & Welling, 2017), MPNN (Gilmer et al., 2017), GAT (Veličković et al., 2018) and GraphSAGE (Hamilton et al., 2017), among others. They encode undirected cyclic graphs into a low-dimensional representation by aggregating and sharing features from neighboring nodes. However, they waste computational resources by repeatedly visiting the nodes when applied to structurally constrained graphs. This led to GNNs for directed acyclic graphs (Thost & Chen, 2021), and for trees, which traverse the graph bottom-up (or up-bottom) and update the nodes only via their children. Examples of tree-GNNs are RNN (Socher et al., 2011; Shuai et al., 2016), Tree-LSTM (Tai et al., 2015) and TreeNet (Cheng et al., 2018).

**Intractable probabilistic models (IPMs).** Extending deep generative models from unstructured to graph-structured data has recently gained significant attention. Variational autoencoders learn a probability distribution over graphs, $p(G)$, by training an encoder and a decoder to map between space of graphs and continuous latent space (Kipf & Welling, 2016; Simonovsky & Komodakis, 2018; Grover et al., 2019). Generative adversarial networks learn $p(G)$ by training (i) a generator to map from latent space to space of graphs and (ii) a discriminator to distinguish whether the graphs

Table 1: *Graph classification.* The test accuracy (higher is better) for the MLP, GRU, LSTM, HMIL, and SPSN networks. It is displayed for the best model in the grid search, which was selected based on the validation accuracy. The results are averaged over 5 runs with different initial conditions. The accuracy is shown with its standard deviation. The average rank is computed as the standard competition ("1224") ranking (Demšar, 2006) on each dataset (lower is better).

| dataset | MLP | GRU | LSTM | HMIL | SPSN |
|---|---|---|---|---|---|
| chess | **0.41±0.03** | **0.41±0.05** | 0.34±0.04 | 0.39±0.02 | 0.39±0.03 |
| citeseer | 0.69±0.02 | 0.74±0.01 | 0.74±0.02 | **0.75±0.01** | **0.75±0.01** |
| cora | 0.75±0.03 | **0.86±0.01** | 0.84±0.01 | 0.85±0.00 | **0.86±0.01** |
| genes | 0.99±0.01 | **1.00±0.01** | 0.98±0.01 | **1.00±0.01** | 0.95±0.01 |
| hepatitis | 0.86±0.02 | **0.88±0.01** | 0.87±0.03 | **0.88±0.02** | **0.88±0.02** |
| mutagenesis | **0.84±0.02** | 0.83±0.02 | 0.82±0.04 | 0.83±0.00 | **0.84±0.02** |
| uwcse | 0.84±0.02 | **0.87±0.03** | 0.85±0.02 | 0.86±0.03 | 0.84±0.02 |
| webkp | 0.77±0.02 | **0.82±0.01** | 0.81±0.02 | **0.82±0.01** | 0.81±0.02 |
| rank | 3.62 | **1.62** | 3.88 | **1.62** | 2.38 |

are synthetic or real (De Cao & Kipf, 2018; Bojchevski et al., 2018). Flow models use the change of variables formula to transform a base distribution on latent space to a distribution on space of graphs, $p(G)$, via an invertible mapping (Liu et al., 2019; Luo et al., 2021). Autoregressive models learn $p(G)$ by using the chain rule of probability to decompose the graph, $G$, into a sequence of subgraphs, constructing $G$ node by node (You et al., 2018; Liao et al., 2019). Diffusion models learn $p(G)$ by noising and denoising trajectories of graphs based on forward and backward diffusion processes, respectively (Jo et al., 2022; Huang et al., 2022; Vignac et al., 2022). NMPs are used in all these generative models, so computing probabilistic (e.g., marginal) queries is intractable.

**Tractable probabilistic models (TPMs).** There has yet to be a substantial interest in probabilistic models facilitating tractable inference for graph-structured data. Graph-structured SPNs (Zheng et al., 2018) decompose cyclic graphs into subgraphs that are isomorphic to a pre-specified set of possibly cyclic templates, designing the conventional SPN for each of them. The sum unit and a layer of the product units aggregate the roots of these SPNs. Graph-induced SPNs (Errica & Niepert, 2023) also decompose cyclic graphs, constructing a collection of trees based on a user-specified neighborhood. The SPNs are not designed for the trees but only for the feature vectors in the nodes. The aggregation is performed by conditioning the sum units at upper levels of the tree by the posterior probabilities at the lower levels. Relational SPNs (RSPNs) (Nath & Domingos, 2015) are TPMs for relational data (a particular form of cyclic graphs). Our set unit is similar to the exchangeable distribution template of the RSPNs. The key difference is that SPSNs model cardinality. The RSPNs do not provide this feature, making them unable to generate new graphs. The mixture densities over finite random sets are most related to SPSNs (Phung & Vo, 2014; Tran et al., 2016; Vo et al., 2018). They can be seen as the sum unit with children given by the set units. These shallow models are designed only for sets. SPSNs generalize them to deep models for hierarchies of sets, achieving higher expressivity by stacking the computational units. SPNs are also used to introduce correlations into graph variational autoencoders (Xia et al., 2023), which are intractable models. Logical circuits can be used to induce probability distributions over discrete objects via knowledge compilation (Chavira & Darwiche, 2008; Ahmed et al., 2022). However, they assume fixed-size inputs, making them applicable only to fixed-size graphs, such as grids.

## 5 EXPERIMENTS

We illustrate the performance and properties of the algebraically tractable SPSN models compared to various intractable, NN-based models. In this context, we would like to investigate their performance in the discriminative learning regime and their robustness to missing values. We provide the implementation of SPSNs at `https://github.com/aicenter/SumProductSet.jl`.

**Models.** To establish the baseline with the intractable models, we choose variants of recurrent NNs (RNNs) for tree-structured data. Though these models are typically used in the NLP domain (Section 1), they are, too, applicable to the tree-structured data in Definition 1. These tree-RNNs differ in the type of cell. We consider the simple multi-layer perceptron (MLP) cell, the gated recurrent unit (GRU) cell (Zhou et al., 2016), and the long-short term memory (LSTM) cell (Tai et al., 2015). The key assumption of these models is that they consider each leaf node to have the same dimension. This requirement does not hold in Definition 1. Therefore, for all these NN-based models, we add a

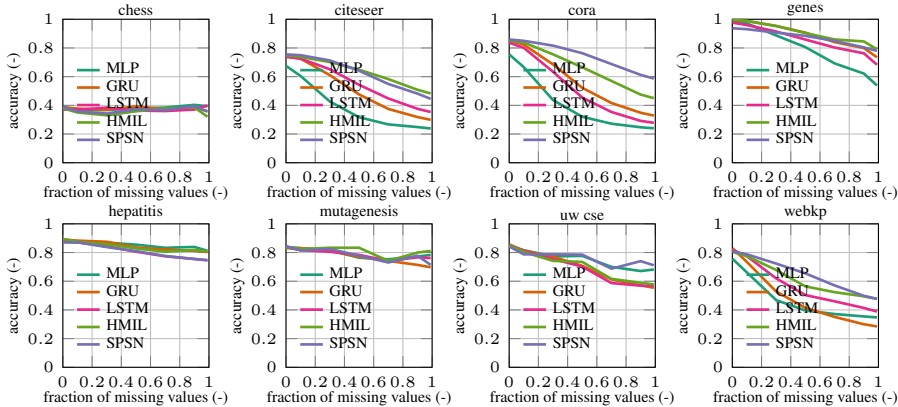

Figure 3: *Missing values.* The test accuracy (higher is better) versus the fraction of missing values for the MLP, GRU, LSTM, HMIL, and SPSN networks. It is displayed for the best model, which was selected based on the validation accuracy. The results are averaged over five runs with different initial conditions.

single dense layer with the linear activation function in front of each leaf node, $v \in L$, to make the input dimension the same. As another competitor, we use the hierarchical multiple-instance learning (HMIL) network (Pevný & Somol, 2016), which is also tailored for the tree-structured data.

**Settings.** We convert eight publicly available datasets from the CTU relational repository (Motl & Schulte, 2015) into the JSON format (Pezoa et al., 2016). The dictionary nodes, list nodes, and atomic nodes of the JSON format directly correspond to the heterogeneous nodes, homogeneous nodes, and leaf nodes of the tree-structured data, respectively (Definition 1, Figure 2). We present the rest of the settings in Section F, including the schemata of the datasets. All models and experiments are implemented in Julia, using `JSONGrinder.jl` and `Mill.jl` (Mandlík et al., 2022).

**Graph classification.** Table 1 shows the test accuracy of classifying the tree-structured graphs. The HMIL and GRU networks deliver the best performance, while the SPSN falls slightly behind. If we look closely at the individual lines, we can see that the SPSN is often very similar to (or the same as) the HMIL and GRU networks. We consider these results unexpectedly good, given that the (NN-based) MLP, GRU, LSTM, and HMIL architectures are denser than the sparse SPSN architecture.

**Missing values.** We consider an experiment where we select the best model in the grid search based on the validation data (as in Table 1) and evaluate its accuracy on the test data containing a fraction of randomly-placed missing values. Figure 3 demonstrates that the SPSN either outperforms or is similar to the NNs. Most notably, for `cora` and `webkp`, the SPSN keeps its classification performance longer compared to the NN models, showing increased robustness to missing values. This experiment applies Proposition 1 to perform marginal inference on the leaf nodes, $v \in L$, that contain missing values. The marginalization is efficient, taking only one pass through the network. Note that the randomness in placing the missing values can lead to situations where all children of the heterogeneous node are missing, allowing us to marginalize the whole heterogeneous node.

## 6    CONCLUSION

We have leveraged the theory of finite random sets to develop a new class of deep learning models—sum-product-set networks (SPSNs)—that represent a probability density over tree-structured graphs. The key advantage of SPSNs is their tractability, which enables exact and efficient inference over specific parts of the data graph. To achieve tractability, SPSNs have to adhere to the structural constraints that are commonly found in other PCs. Consequently, the computational graph of SPSNs has much less connections compared to the computational graph of highly interconnected and non-linear NNs. Notwithstanding this, SPSNs perform comparably to the NNs in the graph classification task, sacrificing only a small amount of performance to retain their tractable properties. Our findings reveal that the tractable and simple inference of SPSNs has also enabled us to achieve results that are comparable to the NNs regarding the robustness to missing values. In future work, we plan to enhance the connectivity within the SPSN block by vectorizing the computational units. We anticipate that this modification will close the small performance gap to the NN models.

ACKNOWLEDGMENTS

The authors acknowledge the support of the GAČR grant no. GA22-32620S and the OP VVV funded project CZ.02.1.01/0.0/0.0/16_019/0000765 "Research Center for Informatics".

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

# A  PROBABILISTIC CIRCUITS

A probabilistic circuit (PC) is a deep learning model representing a joint probability density, $p(\mathbf{x})$, over a fixed-size, *unstructured*, random variable, $\mathbf{x} = (x_1, \ldots, x_n) \in \mathcal{X} \subset \mathbb{R}^n$. The key feature of a PC is that—under certain regularity assumptions—it permits exact and efficient inference scenarios. We define a PC by a parameterized computational graph, $\mathcal{G}$, and a scope function, $\psi$.

**Definition 9.** *(Computational graph).* $\mathcal{G} := (\mathcal{V}, \mathcal{E}, \theta)$ *is a parameterized, directed, acyclic graph, where $\mathcal{V}$ is a set of vertices, $\mathcal{E}$ is set of edges, and $\theta \in \Theta$ are parameters. $\mathcal{V} := (\mathsf{S}, \mathsf{P}, \mathsf{I})$ contains three different subsets of computational units: sum units, $\mathsf{S}$, product units, $\mathsf{P}$, and input units, $\mathsf{I}$. Let $\mathbf{ch}(u)$ and $\mathbf{pa}(u)$ denote the set of child and parent units of $u \in \mathcal{V}$, respectively. If $\mathbf{pa}(u) = \varnothing$, then $u$ is the root unit. If $\mathbf{ch}(u) = \varnothing$, then $u$ is a input unit. We consider that $\mathcal{V}$ contains only a single root unit, and each product unit has only a single parent. The parameters $\theta := (\theta_s, \theta_l)$ are divided into (i) parameters of all sum units, $\theta_u = (w_{u,c})_{c \in \mathbf{ch}(u)}$, which contain non-negative and locally normalized weights (Peharz et al., 2015), $w_{u,c} \geq 0$, $\sum_{c \in \mathbf{ch}(u)} w_{u,c} = 1$; and (ii) parameters of all input units, $\theta_l$, which are specific to a given family of densities, with possibly a different density for each $u \in \mathsf{I}$.*

**Definition 10.** *(Scope function). The mapping $\psi_u : \mathcal{V} \to \mathcal{F}(\mathbf{x})$—from the set of units to the power set of $\mathbf{x}$—outputs a subset of $\mathbf{x} \in \mathcal{X}$ for each $u \in \mathcal{V}$ and is referred to as the scope function. If $u$ is the root unit, then $\psi_u = \mathbf{x}$. If $u$ is a sum unit or a product unit, then $\psi_u = \bigcup_{c \in \mathbf{ch}(u)} \psi_c$.*

PCs are an instance of neural networks (Vergari et al., 2019; Peharz et al., 2020), where each computational unit is a probability density characterized by certain functionality. Input units are the input of a PC. For each $u \in \mathsf{I}$, they compute a (user-specified) probability density, $p_u(\cdot)$, over a subset of $\mathbf{x}$ given by the scope, $\psi_u$, which can be univariate or multivariate (Peharz et al., 2015). Sum units are mixture densities that compute the weighted sum over its children, $p_u(\cdot) = \sum_{c \in \mathbf{ch}(u)} w_{u,c} p_c(\cdot)$, where $w_{u,c}$ (Definition 9) weights the connection between the sum unit and a child unit. Product units are factored densities that compute the product of its children, $p_u(\psi_u) = \prod_{c \in \mathbf{ch}(u)} p_c(\psi_c)$, establishing an unweighted connection between $u$ and $c$ and introducing the conditional independence among the scopes of its children, $\psi_c$. It is commonly the case that (layers of) sum units interleave (layers of) product units. The computations then proceed recursively through $\mathcal{G}$ until reaching the root unit—the output of a PC.

PCs are generally intractable. They instantiate themselves into specific circuits—and thus permit tractability of specific inference scenarios—by imposing various constraints on $\mathcal{G}$, examples include smoothness, decomposability, structured decomposability, determinism, consistency (Chan & Darwiche, 2006; Poon & Domingos, 2011; Shen et al., 2016). In this work, we use only the first two of these constraints, as summarized in Definition 5.

A PC satisfying Definition 5 can be seen as a polynomial composed of input units (Darwiche, 2003). This construction guarantees that any single-dimensional integral interchanges with a sum unit and impacts only a single child of a product unit (Peharz et al., 2015). The integration is then propagated down to the input units, where it can be computed under a closed-form solution (for a tractable form of $u \in \mathsf{I}$). The key practical consequence lies in that various inference tasks—such as integrals of $p(\mathbf{x})$ over $(x_a, \ldots, x_b) \subset \mathbf{x}$—are *tractable* and can be computed in time which is linear in the circuit size (i.e., the cardinality of $\mathcal{V}$). $p(\mathbf{x})$ is guaranteed to be normalized if Definition 5 holds and input units are from the exponential family (Barndorff-Nielsen, 1978). PCs that fulfill Definition 5 are commonly referred to as sum-product networks.

# B  RANDOM FINITE SETS

**Random Finite Sets.** A random finite set (RFS), $X := \{\mathbf{x}_1, \ldots, \mathbf{x}_m\}$, is a random variable taking values in $\mathcal{F}(\mathcal{X})$, the hyperspace of all finite (closed) subsets of some underlying space, $\mathcal{X}$. The randomness of this mathematical object originates from all the elements in the set, $\mathbf{x}_i \in \mathcal{X}$, and also from the cardinality of this set, $m := |X|$, i.e., the number of the elements is itself random. This is the key difference to the standard fixed-size vector, $\mathbf{x} = (x_1, \ldots, x_m)$, where $x_i$'s are stochastic but $m$ is deterministic. The example realizations of an RFS are $X = \varnothing$ (empty set), $X = \{\mathbf{x}_1\}$ (singleton), $X = \{\mathbf{x}_1, \mathbf{x}_2\}$ (tuple), etc. They are points in the hyperspace, $X \in \mathcal{F}(\mathcal{X})$, and each of them is a finite subset of $\mathcal{X}$. The elements of an RFS, $\mathbf{x}_1, \ldots, \mathbf{x}_m$, are assumed distinct (non-

repeated) and unordered. An RFS is then equivalent to a simple point process (Van Lieshout, 2000; Daley et al., 2003; Nguyen, 2006; Mahler, 2007).

A more formal definition of an RFS is as follows. Let $\Omega$ be a sample space, $\sigma(\Omega)$ be a sigma algebra of events on $\Omega$, and $\mathbb{P}$ be a probability measure on the measurable space $(\Omega, \sigma(\Omega))$, i.e., $\mathbb{P}(\Omega) = 1$. Let $\mathcal{X}$ be a locally compact, Hausdorff, separable metric space (e.g., $\mathcal{X} \subseteq \mathbb{R}^n$), $\mathcal{F}(\mathcal{X})$ be the hyperspace of all finite subsets of $\mathcal{X}$, $\sigma(\mathcal{F})$ be a sigma algebra of events on $\mathcal{F}(\mathcal{X})$, and $\mu$ be a dominating (reference) measure on the measurable space $(\mathcal{F}(\mathcal{X}), \sigma(\mathcal{F}))$, which we specify later on. Now consider the probability space and the measure space $(\Omega, \sigma(\Omega), \mathbb{P})$ and $(\mathcal{F}(\mathcal{X}), \sigma(\mathcal{F}), \mu)$, respectively. Then, an RFS is a measurable mapping[2]

$$X : \Omega \to \mathcal{F}(\mathcal{X}). \tag{3}$$

To have the ability to build probabilistic models of the RFS (3), we need tools that characterize its statistical behavior. These tools include a probability distribution, a probability density, and a suitable reference measure to perform the integration. The hyperspace $\mathcal{F}(\mathcal{X})$ does not inherit the standard Euclidean topology, but the Mathéron "hit-or-miss" topology (Mathéron, 1974), which implies that some of these tools are built differently compared to those designed purely for $\mathcal{X}$. However, as demonstrated in this section, we can work with them in a way consistent with the conventional probabilistic calculus.

**Probability distribution.** The probability law of the RFS (3) is characterized by its probability distribution,

$$P(A) \coloneqq \mathbb{P}(X^{-1}(A)) = \mathbb{P}(\{\omega \in \Omega | X(\omega) \in A\}), \tag{4}$$

for any Borel-measurable subset $A \in \sigma(\mathcal{F})$.

**Reference measure.** A measure $\lambda$ on the measurable space $(\mathcal{X}, \sigma(\mathcal{X}))$ is a countably-additive function $\lambda : \sigma(\mathcal{X}) \to [0, \infty]$. It generalizes the notions of length, area, and volume to the subsets of $\mathcal{X}$, which typically involves physical dimensions expressed in the units of $\mathcal{X}$. However, not all measures have units, e.g., the probability measure is unitless. When working with an RFS, one cannot simply use a conventional measure on $(\mathcal{X}, \sigma(\mathcal{X}))$, but it is necessary to extend it to $(\mathcal{F}(\mathcal{X}), \sigma(\mathcal{F}))$. We aim to show how to extend the Lebesgue measure on $(\mathcal{X}, \sigma(\mathcal{X}))$ to a measure on $(\mathcal{F}(\mathcal{X}), \sigma(\mathcal{F}))$. Let $\lambda(S)$ be the Lebesgue measure on $(\mathcal{X}, \sigma(\mathcal{X}))$, for any Borel-measurable subset $S \in \sigma(\mathcal{X})$, and let $\lambda^i(S')$ be the extension of the Lebesgue measure to the Cartesian-product, measurable space $(\mathcal{X}^i, \sigma(\mathcal{X}^i))$, for any subset $S' \in \sigma(\mathcal{X}^i)$. Furthermore, consider a mapping $\chi : \uplus_{i \geq 0} \mathcal{X}^i \to \mathcal{F}(\mathcal{X})$ from vectors of $i$ elements to sets of $i$ elements given by $\chi(\mathbf{x}_1, \ldots, \mathbf{x}_i) = \{\mathbf{x}_1, \ldots, \mathbf{x}_i\}$, where $\uplus$ denotes disjoint union. The mapping $\chi$ is measurable (Goodman et al., 1997; Van Lieshout, 2000), and, therefore, $\chi^{-1}(A)$ is a measurable subset of $\uplus_{i \geq 0} \mathcal{X}^i$ for any subset $A \in \sigma(\mathcal{F})$. Consequently, the reference measure on the measurable space $(\mathcal{F}(\mathcal{X}), \sigma(\mathcal{F}))$—which is commonly adopted in the theory of finite point processes—is defined as follows:

$$\mu(A) = \sum_{i=0}^{\infty} \frac{\lambda^i(\chi^{-1}(A) \cap \mathcal{X}^i)}{c^i i!}, \tag{5}$$

where $\chi^{-1}(A) \cap \mathcal{X}^i \in \sigma(\mathcal{X}^i)$ restricts $\chi^{-1}(A)$ into $i$th Cartesian product of $\mathcal{X}$, respecting the convention $\mathcal{X}^0 \coloneqq \varnothing$. Consider that the unit of measurement in $\mathcal{X}$ is $\iota$, then the unit of measurement of $\lambda^i$ is $\iota^i$. This is why (5) contains the constant $c$ whose unit of measurement is $\iota$. Without this constant, each term in (5) would have different units of measurement, and the infinite sum would be undefined. The measure (5) is therefore unitless.

Say that the units of $p(\mathbf{x}_1)$ are $cm^{-1}$, and the units of $p(\mathbf{x}_1, \mathbf{x}_2)$ are $cm^{-2}$, then it holds that $p(\mathbf{x}_1) > p(\mathbf{x}_1, \mathbf{x}_2)$, see (Vo et al., 2018) for an illustrative example. $c$ thus prevents the incompatibility between the probabilities of two sets with different cardinalities.

**Integral.** The integral of a unitless function $f : \mathcal{F}(\mathcal{X}) \to \mathbb{R}$ over a subset $A \in \sigma(\mathcal{F})$ with respect to the measure $\mu$ is (Geyer, 1999; Mahler, 2007)

$$\int_A f(X)\mu(dX) = \sum_{i=0}^{\infty} \frac{1}{c^i i!} \int_{\chi^{-1}(A) \cap \mathcal{X}^i} f(\{\mathbf{x}_1, \ldots, \mathbf{x}_i\})\lambda^i(d\mathbf{x}_1, \ldots, d\mathbf{x}_i). \tag{6}$$

---

[2]Note the difference to the standard random variable $X : \Omega \to \mathcal{X}$ defined directly on the measure space $(\mathcal{X}, \sigma(\mathcal{X}), \mu)$, where $\mathcal{X}$ is typically equipped with the standard Euclidean topology.

**Remark 1.** *(Tractable integration.) The analytical tractability of* (6) *depends on whether the integral of* $f(\{\mathbf{x}_1, \ldots, \mathbf{x}_i\})$ *w.r.t.* $\lambda^i$ *allows us to find a closed-form solution and whether the infinite sum becomes a finite one. These requirements are satisfied by designing* $f$ *based on a suitable family of functions and ensuring that* $f(\{\mathbf{x}_1, \ldots, \mathbf{x}_i\}) = 0$ *for a sufficiently large* $i$ *(Goodman et al., 1997).*

**Probability density.** The probability density function is the central tool in probabilistic modeling. It is obtained from the Radon-Nikodým theorem (Billingsley, 1995). Its definition states that for two $\sigma$-finite measures $\mu_1$ and $\mu_2$ on the same measurable space $(\mathcal{F}(\mathcal{X}), \sigma(\mathcal{F}))$ there exists an almost everywhere unique function $f : \mathcal{F}(\mathcal{X}) \to [0, \infty)$ such that $\mu_2(A) = \int_A g(X)\mu_1(dX)$ if and only if $\mu_2 << \mu_1$, i.e., $\mu_2$ is absolutely continuous w.r.t. $\mu_1$, or, in other words, $\mu_1(A) = 0$ implies $\mu_2(A) = 0$ for any subset $A \in \sigma(\mathcal{F})$. The function $f = \frac{d\mu_2}{d\mu_1}$ is then referred to as the density function or the Radon-Nikodým derivative of $\mu_2$ w.r.t. $\mu_1$. This allows us to define the probability density function of an RFS as the Radon-Nikodým derivative of the probability measure (4) w.r.t. the reference measure (5),

$$p(X) = \frac{dP}{d\mu}(X), \tag{7}$$

establishing the relation between the two measures as follows: $P(A) = \int_A p(X)\mu(dX)$. The probability density function (7) has no units of measurement since the probability distribution (4) is unitless and the reference measure (5) is also unitless. This contrasts the standard probability density function defined on $\mathcal{X}$, which gives probabilities per unit of $\mathcal{X}$.

**Exchangeability of RFSs.** In point process theory (Daley et al., 2003), the probability density of an RFS (finite point process) is often constructed based on an $m$th-order, non-probabilistic measure, defined on the measurable space $(\mathcal{X}^m, \sigma(\mathcal{X}^m))$, as follows:

$$J_m(A_1, \ldots, A_m) = p(m) \sum_{\text{perm}} P_m(A_{i_1}, \ldots, A_{i_m}), \tag{8}$$

for any $A_i \in \sigma(\mathcal{X})$. Here, $p(m)$ is the cardinality distribution, which determines the total number of elements in the RFS; $P_m$ is the joint distribution on $(\mathcal{X}^m, \sigma(\mathcal{X}^m))$, describing the positions of the elements in the RFS conditionally on $m$; $\sum_{\text{perm}}$ denotes the summation over all $m!$ possible permutations of $i_1, \ldots, i_m$. The measure (8) is *exchangeable* (permutation invariant), i.e., it gives the same value to all permutations of $A_1, \ldots, A_m$. Following this prescription in its full generality would be computationally very expensive, as it requires $m!$ evaluations of $P_m$. Fortunately, we assume that the elements of the RFS follow no specific order, and that we can make a symmetric version of $P_m$ as follows: $P_m^{\text{sym}}(\cdot) = \frac{1}{m!} \sum_{\text{perm}} P_m(\cdot)$, which is simply an equally weighted mixture over all possible permutations. Consequently, after substituting for $\sum_{\text{perm}} P_m$ in (8), we obtain a fully exchangeable—yet computationally more convenient—Janossy measure,

$$J_m(A_1, \ldots, A_m) = p(m)m!P_m^{\text{sym}}(A_1, \ldots, A_m). \tag{9}$$

If (9) is absolutely continuous w.r.t. the reference measure $\lambda^m$, then there exists the Janossy density,

$$j_m(\mathbf{x}_1, \ldots, \mathbf{x}_m) = p(m)m!p_m^{\text{sym}}(\mathbf{x}_1, \ldots, \mathbf{x}_m). \tag{10}$$

Note that (9) and (10) are not a probability measure and a probability density, respectively. Indeed, it holds that $J_m(\mathcal{X}^m) = \int j_m(\mathbf{x}_1, \ldots, \mathbf{x}_m)\lambda^m(d\mathbf{x}_1, \ldots, d\mathbf{x}_m) \neq 1$. However, they are favored for their reduced combinatorial nature and easy interpretability, i.e., $\frac{1}{m!}j_m(\mathbf{x}_1, \ldots, \mathbf{x}_m)\lambda^m(d\mathbf{x}_1, \ldots, d\mathbf{x}_m)$ is the probability of finding exactly one element in each of the $m$ distinct infinitesimal regions. To ensure that (10) is the probability density (7) of an RFS, $X$—which is taken w.r.t. the reference measure (5)—it has to hold that

$$p(\{\mathbf{x}_1, \ldots, \mathbf{x}_m\}) = c^m j_m(\mathbf{x}_1, \ldots, \mathbf{x}_m). \tag{11}$$

**Independent and identically distributed clusters.** The feature (joint) density, $p_m^{\text{sym}}$, in (10) allows us to model the dependencies among the elements of the RFS, $X$. In certain applications, it is more suitable (or simplifying) to assume that the elements, $\mathbf{x}_1, \ldots, \mathbf{x}_m$, are independent and identically distributed (i.i.d.). The feature density then reads $p_m^{\text{sym}}(\mathbf{x}_1, \ldots, \mathbf{x}_m) := \prod_{i=1}^m p(\mathbf{x}_i)$, where $p$ is a probability density on $\mathcal{X}$ indexed by the same parameters for all $i \in \{1, \ldots, m\}$. Note that the assumption of independent elements, but, more importantly, the assumption of identically distributed

elements (the same parameters for all $i \in \{1, \ldots, m\}$), ensures the symmetry of the feature density under all permutations of the elements. The density (11) then becomes

$$p(\{\mathbf{x}_1, \ldots, \mathbf{x}_m\}) = c^m p(m) m! \prod_{i=1}^{n} p(\mathbf{x}_i), \tag{12}$$

which is commonly referred to as the *i.i.d. cluster model*. In the special case, where the cardinality distribution $p(m)$ is the Poisson distribution, (12) represents the Poisson point process (Grimmett & Stirzaker, 2001).

**Remark 2.** *(Independence assumption.) Assumption 1(d) is a simple way to ensure the exchangeability of the set unit. However, it comes at the cost of not capturing the correlations among $\{T_{w_i}\}_{i=i}^{m}$. Despite this fact, we show in Section 5 that the SPSNs deliver solid performance and are very competitive to the NNs. To relax Assumption 1(d), one would need to impose the exchangeability in a different way. For example, a fully general approach would be to use the mixture over $m!$ permutations of $\{T_{w_i}\}_{i=i}^{m}$, as indicated by (8). Since this would be computationally intensive, it is preferable to introduce only approximate exchangeability, which means that one would need to reduce the number of permutations. Albeit such an approach can limit the exchangeability of the set unit to some degree, it does not sacrifice the tractability as long as the components of this mixture form of the feature density are tractable sub-SPSNs.*

## C  EXCHANGEABILITY

**Probabilistic symmetries.** Probabilistic symmetry—the most fundamental one of which is *exchangeability*—is a long-standing subject in the probability literature (Zabell, 2005). The notion of probabilistic symmetry is useful for constructing probabilistic models of exchangeable data structures, including graphs, partitions, and arrays (Orbanz & Roy, 2014). *Infinite* exchangeability is related to the conditionally i.i.d. sequences of random variables via the de Finetti's theorem (de Finetti, 1929; 1937). It states that an infinite sequence of random variables $x_1, x_2, \ldots$ is exchangeable if and only if (iff) there exists a measure $\lambda$ on $\Theta$, such that $p(x_1, \ldots, x_n) = \int \prod_{i=1}^{n} p_\theta(x_i) \lambda(d\theta)$. Consequently, conditionally i.i.d. sequences of random variables are exchangeable. The converse of this assertion is true in the infinite case, $n \to \infty$. *Finite* exchangeability does not satisfy the converse assertion. It defines that for an *extendable* finite sequence[3], $x_1, \ldots, x_n$, the de Finetti's representation holds only approximately, i.e., there is a bounded error between the finite and infinite representations (Diaconis, 1977; Diaconis & Freedman, 1980). We are not interested in finite exchangeability from the perspective of its asymptotic properties. However, we use it to investigate whether a probabilistic model is structurally invariant under the action of a compact group operating on its input, which is considered *non-extendable*.

**Exchangeability of PCs.** As discussed in Section 3.2, the study (and application) of exchangeability in (to) PCs has attracted limited attention. The exchangeability-aware SPNs (Lüdtke et al., 2022) use the mixtures of exchangeable variable models (Niepert & Domingos, 2014; Niepert & Van den Broeck, 2014) as the input units, proposing a structure-learning algorithm that learns the structure by statistically testing the exchangeability within groups of random variables. The relational SPNs (Nath & Domingos, 2015) introduce the exchangeable distribution templates, which are similar in certain aspects to SPSNs. However, though these approaches adopt exchangeable components, none of them investigates how the exchangeability propagates through a PC. Therefore, we characterize the exchangeability of PCs in Proposition 4.

**Proposition 4.** *(Exchangeability of PCs). Let $p(\mathbf{x})$ be a PC satisfying Definition 5 and let $I_E \in I$ be a subset of input units that are exchangeable in the sense of Definition 7 or Definition 8. Then, the PC is partially exchangeable, $p(q_a \cdot \mathbf{x}) = p(\mathbf{x})$, for each $a \in I_E$.*

*Proof.* The result follows from the recursive application of Proposition 2. □

Proposition 4 says that PCs satisfying Definition 5 preserve the exchangeability of their input units. It holds only when there is at least one input unit that is multivariate and exchangeable in the sense

---

[3]This means that the sequence $x_1, \ldots, x_n$ is a part of the longer sequence, $x_1, \ldots, x_m$, $m > n$, with the same statistical properties.

of Definition 7 or Definition 8. Note that the ordering of the scopes (blocks) in the product units remains fixed in the computational graph, i.e., the scopes representing the children of the product units are not exchangeable (only the variables in them).

An alternative way to prove Proposition 4 would be to convert a PC to its mixture representation (Zhao et al., 2016; Trapp et al., 2019). This converted model is a mixture of products of the input units, for which the partial exchangeability can be proven in a way similar to the mixtures of exchangeable variable models (Niepert & Domingos, 2014; Niepert & Van den Broeck, 2014).

## C.1 PROOF OF PROPOSITION 2

*Sum units.* The exchangeability of the sum unit follows from the *smoothness* assumption (Definition 5). The fact that the scope of all children of any sum unit is identical ensures that any permutation (Definition 8), $\boldsymbol{\pi} \in \mathbb{S}_{\mathbf{n}_m}$, propagates through the sum unit, $p_u(\boldsymbol{\pi} \cdot \psi_u) = \sum_{c \in \mathbf{ch}(u)} w_{u,c} p_c(\boldsymbol{\pi} \cdot \psi_u)$, $u \in \mathsf{S}$. In other words, the probability density of the sum unit is partially (or fully) exchangeable, $p_u(\boldsymbol{\pi} \cdot \psi_u) = p_u(\psi_u)$, if and only if the probability densities of all its children are partially (or fully) exchangeable, $p_c(\boldsymbol{\pi} \cdot \psi_u) = p_c(\psi_u)$, for all $c \in \mathbf{ch}(u)$. If we replace $\boldsymbol{\pi}$ by $q_a$, the operator targeting a specific computational unit, we come to the same conclusion.

*Product units.* The exchangeability of the product unit is based on the *decomposability* assumption (Definition 5). The consequence of that the scopes of all children of any product unit are pairwise disjoint is that no matter the type of exchangeability of the child units, the product unit is always only partially exchangeable under the partition of the scopes of its children, $p_u(\boldsymbol{\pi} \cdot \psi_u) = p_u(\pi_1 \cdot \psi_{c_1}, \ldots, \pi_m \cdot \psi_{c_m}) = \prod_{c \in \mathbf{ch}(u)} p_c(\pi_c \cdot \psi_c)$, for all $\boldsymbol{\pi} \in \mathbb{S}_{\mathbf{n}_m}$ and $u \in \mathsf{P}$. Therefore, we can say that the product unit, $p_u(\boldsymbol{\pi} \cdot \psi_u) = p_u(\psi_u)$, preserves the exchangeability of its children.

The product group $\mathbb{S}_{\mathbf{n}_m}$ can be designed such that some of its elements can be an identity group, $\mathbb{S}_{n_i} := \mathbb{I}_{n_i}$. In this case, there exists an identity operator, $e_i$, which does not permute the entries of $\mathbf{x} := (x_1, \ldots, x_{n_i})$, i.e., we have $e_i \cdot \mathbf{x} = (x_1, \ldots, x_{n_i})$. Consequently, $\boldsymbol{\pi} \in \mathbb{S}_{\mathbf{n}_m}$ permutes only *some* of $m$ elements in the collection, $X := (\mathbf{x}_1, \ldots, \mathbf{x}_m)$, e.g., as follows: $\boldsymbol{\pi} \cdot X = (\pi_1 \cdot \mathbf{x}_1, e_2 \cdot \mathbf{x}_2, \ldots, \pi_m \cdot \mathbf{x}_m)$. This allows us to target the permutations only to certain children of the product unit $p_u(\boldsymbol{\pi} \cdot \psi_u) = \prod_{v \in \underline{\mathbf{ch}}(u)} p_v(\pi_v \cdot \psi_v) \prod_{c \in \overline{\mathbf{ch}}(u)} p_c(\psi_c)$, where $\underline{\mathbf{ch}}(u)$ are the children targeted with permutations, and $\overline{\mathbf{ch}}(u)$ are the children that are supposed to stay intact. If we consider replacing $\boldsymbol{\pi}$ by $q_a$, then this principle reveals how to propagate $q_a$ only to those children of the product unit that are the ancestors of $a$. For example, we can have: $q_a \cdot \psi_u = (q_{v_1} \cdot \psi_{v_1}, e_{v_2} \cdot \psi_{v_2}, \ldots, q_{v_m} \cdot \psi_{v_m})$.

*Input units.* The input units are user-specified probability densities, $p_u(\psi_u)$, for each $u \in \mathsf{I}$. The exchangeability of any input unit thus depends on the choice of its density, which can be fully exchangeable (Definition 7), $p_u(\psi_u) = p_u(\boldsymbol{\pi} \cdot \psi_u)$, or partially exchangeable (Definition 8), $p_u(\psi_u) = p_u(\boldsymbol{\pi} \cdot \psi_u)$. This also implies that $q_a = \boldsymbol{\pi}$ if $a = u$. The leaf units terminate the propagation through the computational graph. $\qquad\square$

## C.2 PROOF OF PROPOSITION 3

The result follows from the recursive application of Proposition 2 and the fact that the set unit is fully exchangeable by design (Section B). That is, for any homogeneous node, $T_v := \{T_{w_1}, \ldots, T_{w_m}\}$, it holds that $p_u(\pi \cdot T_v) = p_u(T_v)$ where $v \in O$, $u \in \mathsf{B}$, and $\pi \in \mathbb{S}_m$.

## D TRACTABILITY

The primary purpose of training (learning the parameters of) probabilistic models is to prepare them to answer intricate information-theoretic queries (questions) about events influenced by uncertainty (e.g., computing the probability of some quantities of interest, expectation, entropy). This procedure—referred to as probabilistic inference—often requires calculating integrals of, or w.r.t., the joint probability density representing the model. Many recent probabilistic models deployed in machine learning and artificial intelligence rely on neural networks. The integrals in these models do not admit a closed-form solution, and the inference procedure is, therefore, intractable. To answer even the basic queries with these intractable probabilistic models, we are forced to resort to numerical approximations. The inference procedure is then computationally less efficient, more

complex, and brings more uncertainty into the answers. Tractable probabilistic models, on the other hand, provide a closed-form solution to the integrals involved in the inference procedure and thus answer our queries faithfully to the joint probability density without relying on approximations or heuristics. The inference procedure is then less complicated and computationally more efficient.

**Tractability of PCs.** PCs have become a canonical part of tractable probabilistic modeling. They can answer a range of probabilistic queries *exactly*, i.e., without involving any approximation, and *efficiently*, i.e., in time which is polynomial in the number of edges of their computational graph. The range of admissible probabilistic queries varies depending on the types of structural constraints satisfied by the computational graph (Choi et al., 2020).

We recall only some standard probabilistic queries that are feasible under the usual structural constraints of Definition 5 and can collectively be expressed in terms of the following integral:

$$\lambda(f) = \int f(\mathbf{x}) p(\mathbf{x}) \lambda(d\mathbf{x}). \tag{13}$$

We refer the reader to (Choi et al., 2020; Vergari et al., 2021) for more complex and compositional probabilistic queries.

Even when a PC, $p(\mathbf{x})$, satisfies Definition 5, it does not directly mean that (13) admits a closed-form solution. For this to be the case, the function $f(\mathbf{x})$ has to satisfy certain properties.

**Definition 11.** *(Tractable function for PCs.) Let $f : \mathcal{X} \rightarrow \mathbb{R}$ be a measurable function which factorizes as $f(\mathbf{x}) := \prod_{u \in \mathsf{L}} f_u(\psi_u)$, where $\mathsf{L} \subseteq \mathsf{I}$ is the subset of input units with* unique *and presumably multivariate scopes such that $\mathbf{x} = \bigcup_{u \in \mathsf{L}} \psi_u$. Under this factorization, it follows from the properties of the scope function (Definition 10) that $f_u(\psi_u) := \prod_{c \in \mathbf{ch}(u)} f_c(\psi_c)$ for each $u \in \mathsf{P}$.*

To show how to define various probabilistic queries in terms of the integral (13), we provide examples of $f$. If $f(\mathbf{x}) := \mathbb{1}_A(\mathbf{x})$, where $\mathbb{1}_A$ is the indicator function, and $A := A_1 \times \cdots \times A_m$, then (13) yields $P(A)$, the probability of $A$. Given $A := \mathbf{e}_1 \times \mathbf{e}_2 \times \cdots \times A_{m-1} \times A_m$, where $\mathbf{e}_i$ are evidence assignments, $A_i$ are measurable subsets of $\mathcal{X}$, and $m = |\mathsf{L}|$, we obtain the *marginal query* $P(\mathbf{e}_1, \mathbf{e}_2, \ldots, A_{m-1}, A_m)$, which can easily be used to build a *conditional query* of interest. The *full evidence query* is obtained for $A := \mathbf{e}_1 \times \cdots \times \mathbf{e}_m$. To compute the first-order moment of any $u \in \mathsf{L}$, we define $f_u(\psi_u) := \psi_u$ and $f_a(\psi_a) := 1$ for $a \in \mathsf{L}/(n)$.

**Proposition 5.** *(Tractability of PCs). Let $p(\mathbf{x})$ be a PC satisfying Definition 5 and let $f(\mathbf{x})$ be a function satisfying Definition 11. Then, the integral (13) is tractable and can be computed recursively as follows:*

$$I_u = \begin{cases} \sum_{c \in \mathbf{ch}(u)} w_{u,c} I_c, & \text{for } u \in \mathsf{S}, \\ \prod_{c \in \mathbf{ch}(u)} I_c, & \text{for } u \in \mathsf{P}, \\ \int f_u(\psi_u) p_u(\psi_u) \lambda_u(d\psi_u), & \text{for } u \in \mathsf{I}, \end{cases}$$

*where the measure $\lambda_u(d\psi_u)$ is defined on the space $\Psi_u$, which corresponds to the scope $\psi_u$, and instantiates itself for all $u \in \mathcal{V}$ into either the Lebesgue measure or the counting measure.*

*Proof.* See Section D.1. □

Proposition 5 states that, to compute the integral (13), we have to first compute the resulting values, $I_u$, of the integrals for each input unit, $u \in \mathsf{I}$. Then, $I_u$ is recursively propagated in the feed-forward manner (from the inputs to the root) throughout the computational graph and updated by the sum and product units.

## D.1 Proof of Proposition 5

We aim to demonstrate that the integral (13) is tractable. We show this in a recursive manner, considering how the integral propagates through each computational unit of $p(\mathbf{x})$.

Before we start, let us remind Definition 11, which shows that $f(\mathbf{x}) := \prod_{i \in \mathsf{L}} f_i(\psi_i)$. Considering Definition 5 is satisfied; then, based on Definition 10, the partial factorization $f_u(\psi_u) := \prod_{i \in \mathsf{L}_u} f_i(\psi_i)$ can be extracted from $f(\mathbf{x})$ for any $u \in (\mathsf{S}, \mathsf{P})$. Here, $\mathsf{L}_u \subseteq \mathsf{L}$ contains only the input

units that are reachable from $u$ and have a unique scope. The partial factorization, $f_u(\psi_u)$, leads to the definition of the following *intermediate* integral:

$$\lambda_u(f_u) = \int f_u(\psi_u) p_u(\psi_u) \lambda_u(d\psi_u), \tag{14}$$

which acts on a given computational unit $u$. In (14), $p_u(\psi_u)$ is the probability density of $u$, and $\lambda_u$ is the reference measure on the measurable space $(\Psi_u, \sigma(\Psi_u))$. We will see that the proof consists of seeking an algebraic closure (recursion) for the functional form of the integral (14).

*Input units.* The existence of a closed-form solution of the integral (14) for $u \in \mathsf{I}$ is ensured if $p_u(\psi_u)$ is selected from a family of tractable probability densities (e.g., the exponential family (Barndorff-Nielsen, 1978)) and $f_u(\psi_u)$ is an algebraically simple function that does not prevent the solution to be found. The solution, therefore, depends purely on our choice. We use $I_u$ to denote a concrete value of the integral.

*Sum units.* Consider that the smoothness assumption (Definition 5) is satisfied. Then, after substituting the probability density of the sum unit for $p_u(\psi_u)$ into (14), and exchanging the integration and summation utilizing the Fubini's theorem (Weir, 1973), we obtain

$$\int f_u(\psi_u) p_u(\psi_u) \lambda_u(d\psi_u) = \sum_{c \in \mathbf{ch}(u)} w_{u,c} \int f_u(\psi_u) p_c(\psi_u) \lambda_u(d\psi_u),$$

where $u \in \mathsf{S}$. The smoothness assumption (Definition 5) states that the children of the sum unit have an identical scope. It implies that the integration affects each child of the sum unit in the same way. Consequently, the integral w.r.t. $p_u$ can be solved if the integrals w.r.t. $p_c$ can be solved for all $c \in \mathbf{ch}(u)$. In other words, the sum unit is tractable if all its children (i.e., other sub-PCs) are tractable. The tractability of the sum unit thus propagates from its children. We can see that the functional form of the l.h.s. integral is the same as the functional form of the r.h.s. integrals representing the children of the sum unit. Therefore, we can replace these functional prescriptions with concrete realizations $I_u$ (l.h.s.) and $I_c$ (r.h.s.). The sum unit then propagates already realized values of these integrals and multiplies them by the weights.

*Product units.* Assume that the decomposability assumption (Definition 5) holds. Then, after substituting the probability density of the product unit for $p_u(\psi_u)$ into (14), and factorizing the function $f_u(\psi_u)$ in accordance with the pairwise disjoint scopes of the product unit, i.e., $f_u(\psi_u) := \prod_{c \in \mathbf{ch}(u)} f_c(\psi_c)$, we have

$$\int f_u(\psi_u) p_u(\psi_u) \lambda_u(d\psi_u) = \int \prod_{c \in \mathbf{ch}(u)} f_c(\psi_c) \prod_{c \in \mathbf{ch}(u)} p_c(\psi_c) \lambda_c(d\psi_c)$$

$$= \prod_{c \in \mathbf{ch}(u)} \int f_c(\psi_c) p_c(\psi_c) \lambda_c(d\psi_c),$$

where $u \in \mathsf{P}$. The decomposability assumption (Definition 5) says that children of the product unit have independent scopes. The consequence is that the integral reduces to the product of simpler integrals. The integral w.r.t. $p_u$ is tractable if the integrals of all its children $p_c$ are tractable for all $c \in \mathbf{ch}(n)$, i.e., the tractability of the product unit propagates from its children. Similarly as before, the functional forms of the l.h.s. integral of the product unit and the r.h.s. integrals of its children are the same, which allows us to replace them with concrete realizations $I_u$ (l.h.s.) and $I_c$ (r.h.s.).

If the tractability holds for each unit $u \in \{\mathsf{S}, \mathsf{P}, \mathsf{I}\}$ in the computational graph, then a PC is tractable and (13) admits a closed-form solution. □

## D.2 PROOF OF PROPOSITION 1

Our goal is to show that the integral (2) can be computed recursively under a closed-form solution. To this end, we proceed analogously as in the proof of Proposition 5. Recall from Definition 6 that $f(T) := \prod_{i \in \mathsf{L}} f_i(\psi_i)$, and if Definition 5 and Assumption 1 hold, then it follows from the properties of Definition 4 that the partial factorization $f_u(\psi_u) := \prod_{i \in \mathsf{L}_u} f_i(\psi_i)$ can be extracted from $f(T)$ for any $u \in \{\mathsf{S}, \mathsf{P}, \mathsf{B}\}$, where $\mathsf{L}_u \subseteq \mathsf{L}$ is the set of the input units that can be reached from $u$ and have

the unique scope. Similarly as before, the existence of $f_u(\psi_u)$ allows us to define the *intermediate* integral, which acts on a given computational unit $u$, as follows:

$$\nu_u(f_u) = \int f_u(\psi_u) p_u(\psi_u) \nu_u(d\psi_u), \qquad (15)$$

where $p_u(\psi_u)$ is the probability density of $u$, and $\nu_u$ is the reference measure on the measurable space $(\Psi_u, \sigma(\Psi_u))$. $\Psi_u$ can take various forms depending on whether the scope $\psi_u$ is the homogeneous node, (a part of) the heterogeneous node or the leaf node. If $\psi_u$ is the homogeneous node, then $\Psi_u$ is the hyperspace of all finite subsets of some underlying space, $\Psi$, which characterizes the feature density of the set unit. If $\psi_u$ is the heterogeneous node, then $\Psi_u$ is the Cartesian product space composed of, e.g., continuous spaces, discrete spaces, but also hyperspaces defining other RFSs. If $\psi_u$ is the leaf node, then $\Psi_u$ can be the Cartesian product of continuous and (or) discrete spaces. For this reason, the reference measure $\nu_u$ instantiates itself depending on a given computational unit, $u$, and can take various forms based on the space, $\Psi_u$.

In the case $\psi_u$ is a (subset of) leaf node(s), the proof is carried out in the same way as for Proposition 5. This is also true when $\psi_u$ is a (subset of) heterogeneous node(s). The difference is that $\lambda_u$ in (14) is replaced by a more general measure $\nu_u$. This leaves us to prove only the last case where $\psi_u$ is a homogeneous node.

*Set units.* Consider the scope of the set unit, $\psi_u$, $u \in B$, is an RFS taking values in $\Psi_u := \mathcal{F}(\Psi_w)$, the hyperspace of all finite subsets of some underlying space, $\Psi_w$. That is, we have $\psi_u = \{\psi_{w_1} \ldots, \psi_{w_k}\}$, where $\psi_{w_i} \in \Psi_w$ are $k$ distinct instances of the *identical* scope of the feature density of the set unit $u$. Furthermore, let $\nu_u$ be the unitless reference measure (5) on $(\Psi_u, \sigma(\Psi_u))$, associated to the set unit, $p_u(\psi_u)$, and let $\nu_w$ be the reference measure on $(\Psi_w, \sigma(\Psi_w))$, corresponding to the feature density, $p(\psi_{w_i})$. Now, from the properties of $f$, we have $f_u(\psi_u) = \prod_{i=1}^k f_i(\psi_{w_i})$, and, after substituting this function, along with the density of the set unit (1), into (15), we obtain

$$\int f_u(\psi_u) p_u(\psi_u) \nu_u(d\psi_u)$$

$$= \sum_{k=0}^\infty \frac{1}{c^k k!} \int_{\Psi_w^k} f_u(\{\psi_{w_1}, \ldots, \psi_{w_k}\}) p_u(\{\psi_{w_1}, \ldots, \psi_{w_k}\}) \nu_w^k(d\psi_{w_1}, \ldots, d\psi_{w_k})$$

$$= \sum_{k=0}^\infty \frac{1}{c^k k!} \int_{\Psi_w^k} \prod_{i=1}^k f_i(\psi_{w_i}) c^k k! p(k) \prod_{i=1}^k p(\psi_{w_i}) \nu_w(d\psi_{w_i})$$

$$= \sum_{k=0}^\infty p(k) \prod_{i=1}^k \int_{\Psi_w} f_i(\psi_{w_i}) p(\psi_{w_i}) \nu_w(d\psi_{w_i})$$

$$= \sum_{k=0}^\infty p(k) \prod_{i=1}^k \int f_i(\psi_{w_i}) p(\psi_{w_i}) \nu_w(d\psi_{w_i}),$$

where $\Psi_w^k := \Psi_w \times \cdots \times \Psi_w$. Once again, we can see that the functional form of the l.h.s. integral is the same as that of the r.h.s. integrals, allowing us to replace them with $I_u$ and $I_i$. $\qquad \square$

# E   IMPLEMENTATION

There are multiple ways to implement SPSNs. However, the construction of the computational graph always has to follow from the properties of computational units (Definition 3) and respect the structural constraints (Definition 5). We provide more details on the *layer-wise* implementation introduced in Section 3. However, before that, we first present an intuitive description of constructing SPSNs in a *node-wise* manner.

## E.1   NODE-WISE APPROACH

Let us consider the example in Figure 4. The left part (Figure 4(a)) shows the data graph (Definition 1) and its schema (Definition 2) highlighted by the dashed line. We can see that the schema is in fact a simplified graph that excludes the structurally identical children of the homogeneous nodes

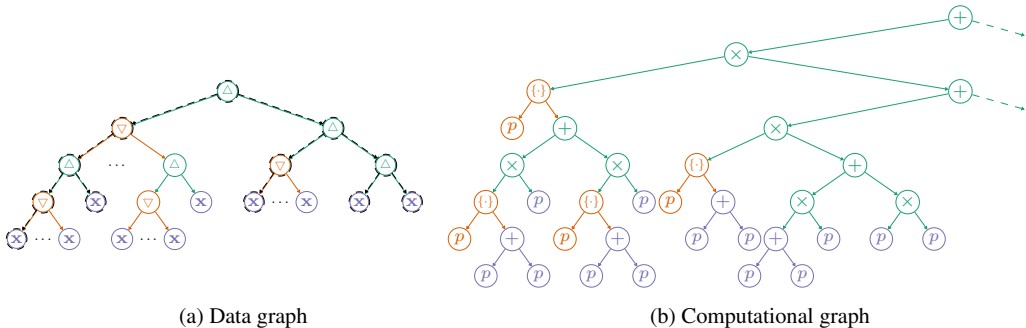

(a) Data graph  (b) Computational graph

Figure 4: *Sum-product-set networks.* (a) The tree-structured, heterogeneous, data graph, $T$, (Definition 1), and the *schema* (dashed line), $S$, (Definition 2). Here, $\triangle$ is the heterogeneous node, $\triangledown$ is the homogeneous node, and $\mathbf{x}$ denotes the leaf node of $T$. (b) The computational graph, $\mathcal{G}$, of an SPSN (Definition 3) designed based on $S$, where $+$, $\times$, $\{\cdot\}$ and $p$ are the sum unit, product unit, *set* unit and leaf unit of $\mathcal{G}$, respectively. The subtrees of $T$ in (a) are modeled by the corresponding parts of $\mathcal{G}$ in (b), as displayed in green, orange, and blue. The dashed arrow in (b) (the second child in the top sum units) represents the same sub-network as in the first child. In this example, we consider $n_l = 1$, $n_s = 2$, and $n_p = 2$.

and keeps only the child that allows us to reach the deepest level of the tree. The construction of an SPSN follows from the schema. We start at the root heterogeneous node. Any heterogeneous node can be modeled by possibly many alterations of sum units and product unis. It is possible to recursively split the heterogeneous node as long as it still contains enough elements since, every time we apply the product unit, we split the heterogeneous node into two parts (or more depending on $n_p$). The right part (Figure 4(b)) displays that the root heterogeneous node is modeled by only a single sum unit whose children are two product units (the right one is indicated by the dashed arrow, which we hide for simplicity). This is due to the fact that the root heterogeneous node contains only two children and the product unit separated the children into two singletons: homogeneous node and heterogeneous node depicted in the left and right child of the root heterogeneous node of the data graph. The homogeneous node can be modeled only by the set units; therefore, we add a set unit into the first child of the aforementioned product unit. The heterogeneous node has again only two children, which means that we will model it in the same way as the root one, i.e., by using a single sum unit with a product unit in each of its children. Now, if we go back to the homogeneous node, then we can see that, in the schema, its child is another heterogeneous node with two children. Therefore, we repeat the same process as before. Since one of the children of this heterogeneous node is the leaf node, we place an input unit into the computational graph. This continues until we traverse all parts of the schema, extending the computational graph in the process.

## E.2  LAYER-WISE APPROACH

The node-wise approach is a simple mechanism to construct SPSNs. Nonetheless, it is often computationally inefficient for the implementation with modern automatic differentiation tools. The reason for this lies in that these tools have to produce their own computational diagram (graph) from the computational graph of the SPSN. Therefore, we provide a layer-wise approach to construct SPSNs, simplifying the underlying differentiation mechanisms.

Algorithm 1 contains the procedure $\texttt{spsn\_network}(\psi, n_c, n_l, n_s, n_p)$ which constructs an SPSN based on the following inputs: $\psi$ is the scope of the root unit (which we set to the schema when applying the procedure), $n_c$ is the number of root units of the network (we set $n_c > 1$ when using the network for classification), and $n_l$, $n_s$, and $n_p$ are the number of layers, children in the sum units, and children of the product units, respectively, which are common to all blocks in the network. This imposes a regular structure on the network and makes it suitable to the layer-wise ordering of the computational units. The resulting computational graph is consequently more efficient for the implementation with the automatic differentiation tools and also more convenient for parallelization on the contemporary computational hardware.

The key procedure is $\texttt{spsn\_block}$, which recursively constructs the computational graph of an SPSN in the block-by-block manner as depicted in Figure 2. We first create an empty block (line 1). Then,

we continue by creating layers of scope functions in `scope_layers` (line 2). We provide more details on this procedure below. For each layer of these scopes, we add (via ←) two layers of computational units into the block. (i) The layer of sum units, `slayer`, where the first argument is the number of children of each sum unit and the second argument is the number of sum units. (ii) The layer of product units, `player`, where the meaning of the arguments is the same as with `slayer`. We repeat this until $n_l = 1$. Then, we add the layer of input units, `ilayer`. This layer assigns an input unit to each scope in $l$ based on its type. That is, if the scope represents the leaf nodes, then it checks the type of data (floats, integers, strings) and creates appropriate probability density. If the scope is the homogeneous node, then it creates the set unit. The feature densities of all set units in this input layer are not connected to any other part of the computational graph at this moment. The procedure then proceeds by gathering all these unconnected scopes via `scope_set_units`. All these steps are now repeated by calling `spsn_block` again (line 14). Every iteration, `spsn_block` returns the network that was created to this moment, $N$, and the scopes of the unplugged set units that are provided to the next iteration. The network that has been generated so far, $N$, is connected to the unplugged scopes of the set units in the current block $B$ by using `connect`.

Algorithm 2 presents the `scope_layers` procedure. It starts by assigning the input set of scopes $\psi$ into $L$, a structure holding all layers of scopes, creating the first layer of scopes. The next layer is made by the `scope_slayer` and `scope_player` procedures. `scope_slayer` uses `repeat` to make $n_s$ copies of each element in $\psi$, to reflect the fact that the children of the sum unit have the identical scope. Similarly, `scope_player` splits each element in $\psi$ into $n_p$ parts, to reflect the fact that the children of the product unit have disjoint scopes. This process is repeated until we either (i) reach maximum allowable number of layers or (ii) there is a scope in $\psi$ represented by a singleton. The latter is realized by `minimum_length`, which first evaluates the number of elements in each scope of $\psi$ and then finds their minimum.

**The block size.** The regular structure of the SPSN block allows us to find a closed-form solution for its size in terms of the number of computational units containing parameters. The number of sum units is given by $K_s = |\psi| \sum_{l=0}^{n_l-1} (n_s n_p)^l$, where $|\psi|$ is the cardinality of the input set of scopes in `spsn_block`. The number of input units is $K_i = |\psi|(n_s n_p)^l$. While $K_s$ can directly be used to compute the number of parameters in all sum units, $K_i$ serves only to complete an intuition about the size of each block. To obtain a concrete number of parameters in the input layer, we need to count the parameters in each of its units due to the differences in the data types.

**Algorithm 1** Construct the SPSN network

**procedure** spsn_network($\psi, n_c, n_l, n_s, n_p$)
1: $\psi = \texttt{repeat}(\psi, n_c)$
2: $N, \_ = \texttt{spsn\_block}(\psi, n_l, n_s, n_p)$
3: **return** $N$

**procedure** spsn_block($\psi, n_l, n_s, n_p$)
1: $B = \varnothing$
2: $L = \texttt{scope\_layers}(\psi, n_l, n_s, n_p)$
3: **for all** $l \in L$ **do**
4:    $k = \texttt{length}(l)$
5:    **if** $n_l > 1$ **then**
6:      $B \leftarrow \texttt{slayer}(n_s, k)$
7:      $B \leftarrow \texttt{player}(n_p, k * n_s)$
8:    **else**
9:      $B \leftarrow \texttt{ilayer}(l)$
10:      $\psi = \texttt{scope\_set\_units}(l)$
11:    **end if**
12:    $n_l = n_l - 1$
13: **end for**
14: $N, \psi = \texttt{spsn\_block}(\psi, n_l, n_s, n_p)$
15: $N = \texttt{connect}(N, B)$
16: **return** $N, \psi$

**Algorithm 2** Procedures

**procedure** scope_layers($\psi, n_l, n_s, n_p$)
1: $L = (\psi)$
2: **while** true **do**
3:    $\psi = \texttt{scope\_slayer}(\psi, n_s)$
4:    $\psi = \texttt{scope\_player}(\psi, n_p)$
5:    $L \leftarrow \psi$
6:    **if** minimum_lenght($\psi$) **is** 1 **or** $n_l$ **is** 1 **then**
7:      **break**
8:    **else**
9:      $n_l = n_l - 1$
10:    **end if**
11: **end while**
12: **return** $L$

**procedure** scope_slayer($\psi, n_s$)
1: $\bar{\psi} = \varnothing$
2: **for all** $s \in \psi$ **do**
3:    $\bar{\psi} \leftarrow \texttt{repeat}(s, n_s)$
4: **end for**
5: **return** $\bar{\psi}$

**procedure** scope_player($\psi, n_p$)
1: $\bar{\psi} = \varnothing$
2: **for all** $s \in \psi$ **do**
3:    $\bar{\psi} \leftarrow \texttt{split}(s, n_p)$
4: **end for**
5: **return** $\psi$

## F  EXPERIMENTAL SETTINGS

The leaf nodes, $v \in L$, contain different data types, including reals, integers, and strings. We use the default feature extractor from `JSONGrinder.jl` (v2.3.2) to pre-process these data. We perform the grid search over the hyper-parameters of the models mentioned in Section 5. For the MLP, GRU, and LSTM networks, we set the dimension of the hidden state(s) and the output in $\{10, 20, 30, 40\}$. For the HMIL network, we use the default settings of the model builder from `Mill.jl` (v2.8.1), only changing the number of hidden units of all the inner layers in $\{10, 20, 30, 40\}$. We add a single dense layer with the linear activation function to adapt the outputs of these networks to the number of classes in the datasets. For the SPSN networks, we choose the Poisson distribution as the cardinality distribution and the following hyper-parameters: $n_l \in \{1, 2, 3\}$, $n_s \in \{2, 3, \ldots, 10\}$, and $n_p := 2$. We use the ADAM optimizer (Kingma & Ba, 2014) with fixing 10 samples in the minibatch and varying the step-size in $\{0.1, 0.01, 0.001\}$. The datasets are randomly split into 64%, 16%, and 20% for training, validation, and testing, respectively.

We performed the experiments on a computational cluster equipped with 116 CPUs (Intel Xeon Scalable Gold 6146). The jobs to perform the grid search over the admissible range of hyper-parameters were scheduled by SLURM 23.02.2. We limited each job to a single core and 8GB of memory. The computational time was restricted to one day, but all jobs were finished under that limit (ranging approximately between 2-18 hours per dataset).

## G  DATASETS

The CTU Prague relational learning repository (Motl & Schulte, 2015) is a rich source of structured data. These data form a directed graph where the nodes are tables and edges are the foreign keys. Some of these datasets are already in the form of threes; however, there are also graphs containing cycles. As a part of the preprocessing, we decompose these cyclic graphs to tree graphs by selecting a node and then reaching to the neighborhood nodes in the one-hop distance.

Table 2 shows the two main attributes of the datasets under study: the number of instances (i.e., the number of tree-structured graphs) and the number of classes of these instances. A detailed description of these datasets, additional attributes, and accompanying references are accessible at `https://relational.fit.cvut.cz/`.

Figure 1 shows a single instance of the tree-structured graph data in the JSON format (Pezoa et al., 2016), and Figure 5 illustrates the corresponding schema (Definition 2 of the main paper). As can be seen (and as also mentioned in the main paper), the leaf nodes contain different data types: integers, floats, and strings. In Figures 6-12, we provide the schemata of the remaining datasets in Table 2.

Table 2: *Datasets.* The number of instances (trees) and the number of classes of the datasets under study. $n_O$, $n_H$, and $n_L$ are the numbers of homogeneous nodes, heterogeneous nodes, and leaf nodes, respectively, in the whole dataset. $\bar{n}_O$, $\bar{n}_H$, and $\bar{n}_L$ show the corresponding average number of nodes per tree. The last column is the size of the dataset in MiBs. **For comparison, the full training MNIST dataset has around 45 MiBs.**

| dataset | # of instances (trees) | # of classes | $n_O$ | $n_H$ | $n_L$ | $\bar{n}_O$ | $\bar{n}_H$ | $\bar{n}_L$ | size [MiB] |
|---|---|---|---|---|---|---|---|---|---|
| mutagenesis | 188 | 2 | 5081 | 15567 | 57375 | 27 | 83 | 305 | 3.3 |
| genes | 862 | 15 | 3544 | 17941 | 125712 | 4 | 21 | 146 | 7.7 |
| cora | 2708 | 7 | 16274 | 13566 | 247663 | 6 | 5 | 91 | 9.3 |
| citeseer | 3312 | 6 | 16071 | 12759 | 411929 | 5 | 4 | 124 | 14.4 |
| webkp | 877 | 5 | 4970 | 4093 | 396890 | 6 | 5 | 453 | 13.2 |
| chess | 295 | 3 | 10325 | 590 | 53593 | 35 | 2 | 182 | 1.8 |
| uw_cse | 278 | 4 | 782 | 782 | 3128 | 3 | 3 | 11 | 0.170 |
| hepatitis | 500 | 2 | 1500 | 7008 | 65039 | 3 | 14 | 130 | 2.3 |

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

├── lumo: [Float64]  # n_unique = 177, n_inst = 188
├── inda: [Int64]  # n_unique = 2, n_inst = 188
├── logp: [Float64,Int64]  # n_unique = 107, n_inst = 188
├── ind1: [Int64]  # n_unique = 2, n_inst = 188
└── atoms: [List]  # n_inst = 188
         └── [Dict]  # n_inst = 4893
                  ├── element: [String]  # n_unique = 7, n_inst = 4893
                  ├── bonds: [List]  # n_inst = 4893
                  │        └── [Dict]  # n_inst = 10486
                  │                 ├── element: [String]  # n_unique = 7, n_inst = 10486
                  │                 ├── type_bond: [Int64]  # n_unique 6, n_inst = 10486
                  │                 ├── type_atom: [Int64]  # n_unique = 36, n_inst = 10486
                  │                 └── charge: [Float64]  # n_unique = 444, n_inst = 10486
                  ├── type_atom: [Int64]  # n_unique = 36, n_inst = 4893
                  └── charge: [Float64]  # n_unique = 444, n_inst = 4893
```

Figure 5: *Schema.* The schema of the `mutagenesis` dataset.

```
[Dict]  # n_inst = 862
  ├── interactions: [List]  # n_inst = 862
  │                  └── [Dict]  # n_inst = 1820
  │                              ├── type: [String]  # n_unique = 3, n_inst = 1820
  │                              ├── expression_Corr: [Float64,Int64]  # n_unique = 817, n_inst = 1820
  │                              └── records: [List]  # n_inst = 1820
  │                                          └── [Dict]  # n_inst = 10913
  │                                                      ├── localization: [String]
  │                                                      │              # n_unique = 13, n_inst = 10913
  │                                                      ├── complex: [String]
  │                                                      │         # n_unique = 44, n_inst = 10913
  │                                                      ├── chromosome: [Int64]
  │                                                      │            # n_unique = 17, n_inst = 10913
  │                                                      ├── function: [String]
  │                                                      │          # n_unique = 13, n_inst = 10913
  │                                                      ├── essential: [String]
  │                                                      │           # n_unique = 4, n_inst = 10913
  │                                                      ├── class: [String]
  │                                                      │       # n_unique = 22, n_inst = 10913
  │                                                      ├── phenotype: [String]
  │                                                      │           # n_unique = 13, n_inst = 10913
  │                                                      └── motif: [String]
  │                                                              # n_unique = 183, n_inst = 10913
  └── records: [List]  # n_inst = 862
              └── [Dict]  # n_inst = 4346
                          ├── localization: [String]  # n_unique = 15, n_inst = 4346
                          ├── complex: [String]  # n_unique = 52, n_inst = 4346
                          ├── chromosome: [Int64]  # n_unique = 17, n_inst = 4346
                          ├── function: [String]  # n_unique = 13, n_inst = 4346
                          ├── essential: [String]  # n_unique = 4, n_inst = 4346
                          ├── class: [String]  # n_unique = 24, n_inst = 4346
                          ├── phenotype: [String]  # n_unique = 13, n_inst = 4346
                          └── motif: [String]  # n_unique = 236, n_inst = 4346
```

Figure 6: The schema of the genes dataset.

```
[Dict]  # n_inst = 2708
  ├── citing: [List]  # n_inst = 2708
  │          └── [Dict]  # n_inst = 10858
  │                      └── word_cited_id: [List]  # n_inst = 10858
  │                                        └── [String]  # n_unique = 1432, n_inst = 198447
  └── word_cited_id: [List]  # n_inst = 2708
                     └── [String]  # n_unique = 1432, n_inst = 49216
```

Figure 7: The schema of the cora dataset.

```
[Dict]  # n_inst = 3312
  ├── citing: [List]  # n_inst = 3312
  │          └── [Dict]  # n_inst = 9447
  │                      └── word_cited_id: [List]  # n_inst = 9447
  │                                        └── [String]  # n_unique = 3703, n_inst = 306764
  └── word_cited_id: [List]  # n_inst = 3312
                     └── [String]  # n_unique = 3703, n_inst = 105165
```

Figure 8: The schema of the citeseer dataset.

```
[Dict]  # n_inst = 877
  ├── citing: [List]  # n_inst = 877
  │          └── [Dict]  # n_inst = 3216
  │                      └── word_cited_id: [List]  # n_inst = 3216
  │                                        └── [String]  # n_unique = 1703, n_inst = 317525
  └── word_cited_id: [List]  # n_inst = 877
                     └── [String]  # n_unique = 1703, n_inst = 79365
```

Figure 9: The schema of the webkp dataset.

```
[Dict]   # n_inst = 278
├──────────── person: [Dict]   # n_inst = 278
│                      ├──── hasPosition: [String]   # n_unique = 5, n_inst = 278
│                      ├──────── student: [String]   # n_unique = 2, n_inst = 278
│                      ├──────── professor: [String]   # n_unique = 2, n_inst = 278
│                      ├──────────── id: [Int64]   # n_unique = 278, n_inst = 278
│                      ├──── yearsInProgram: [String]   # n_unique = 12, n_inst = 278
│                      ├──────── courses: [List]   # n_inst = 278
│                                         └── [String]   # n_unique = 3, n_inst = 189
└──── interactions: [List]   # n_inst = 278
                     └── [Dict]   # n_inst = 226
                                ├──────── hasPosition: [String]   # n_unique = 5, n_inst = 226
                                ├──────── student: [String]   # n_unique = 2, n_inst = 226
                                ├──────── professor: [String]   # n_unique = 2, n_inst = 226
                                ├──────────── id: [Int64]   # n_unique = 130, n_inst = 226
                                ├──── yearsInProgram: [String]   # n_unique = 12, n_inst = 226
                                ├──────── courses: [List]   # n_inst = 226
                                                   └── [String]   # n_unique = 3, n_inst = 419
```

Figure 10: The schema of the the `uw_cse` dataset.

```
[Dict]   # n_inst = 500
├──── sex: [String]   # n_unique = 2, n_inst = 500
├──── age: [String]   # n_unique = 7, n_inst = 500
├──── inf: [List]   # n_inst = 500
│            └── [Dict]   # n_inst = 196
│                       └── dur: [String]   # n_unique = 5, n_inst = 196
├──── bio: [List]   # n_inst = 500
│            └── [Dict]   # n_inst = 621
│                       ├── activity: [String]   # n_unique = 5, n_inst = 621
│                       └── fibros: [String]   # n_unique = 5, n_inst = 621
└──── indis: [List]   # n_inst = 500
             └── [Dict]   # n_inst = 5691
                        ├──── dbil: [String]   # n_unique = 2, n_inst = 5691
                        ├──── tcho: [String]   # n_unique = 4, n_inst = 5691
                        ├──── gpt: [String]   # n_unique = 4, n_inst = 5691
                        ├──── alb: [String]   # n_unique = 2, n_inst = 5691
                        ├──── tp: [String]   # n_unique = 4, n_inst = 5691
                        ├──── ttt: [String]   # n_unique = 6, n_inst = 5691
                        ├──── got: [String]   # n_unique = 5, n_inst = 5691
                        ├──── che: [String]   # n_unique = 10, n_inst = 5691
                        ├── in_id: [Int64]   # n_unique = 5691, n_inst = 5691
                        ├──── ztt: [String]   # n_unique = 6, n_inst = 5691
                        └──── tbil: [String]   # n_unique = 2, n_inst = 5691
```

Figure 11: The schema of the `hepatitis` dataset.

```
[Dict]  # n_inst = 295
├─────────── w3: [List]  # n_inst = 295
│                └── [Int64]  # n_unique = 16, n_inst = 793
├─────────── w7: [List]  # n_inst = 295
│                └── [Int64]  # n_unique = 23, n_inst = 866
├─────────── b5: [List]  # n_inst = 295
│                └── [Int64]  # n_unique = 23, n_inst = 836
├─────────── b2: [List]  # n_inst = 295
│                └── [Int64]  # n_unique = 15, n_inst = 678
├────────── white: [List]  # n_inst = 295
│                └── [Int64]  #  n_unique = 54, n_inst = 4855
├─────────── w6: [List]  # n_inst = 295
│                └── [Int64]  # n_unique = 23, n_inst = 857
├─────────── w4: [List]  # n_inst = 295
│                └── [Int64]  # n_unique = 22, n_inst = 839
├─────────── b8: [List]  # n_inst = 295
│                └── [Int64]  # n_unique = 24, n_inst = 860
├────────── event: [List]  # n_inst = 295
│                └── [Int64]  # n_unique = 18, n_inst = 6195
├─────────── b9: [List]  # n_inst = 295
│                └── [Int64]  # n_unique = 25, n_inst = 897
├──── whiteElo: [Int64]  # n_unique = 88, n_inst = 295
├── event_date: [String]  # n_unique = 2, n_inst = 295
├─────────── b1: [List]  # n_inst = 295
│                └── [Int64]  # n_unique = 8, n_inst = 681
├─────────── w1: [List]  # n_inst = 295
│                └── [Int64]  # n_unique = 8, n_inst = 620
├─────────── b6: [List]  # n_inst = 295
│                └── [Int64]  # n_unique = 23, n_inst = 828
├────────── site: [List]   # n_inst = 295
│                └── [Int64]  # n_unique = 9, n_inst = 2655
├─────────── w5: [List]  # n_inst = 295
│                └── [Int64]  # n_unique = 23, n_inst = 830
├────────── ECO: [List]  # n_inst = 295
│                └── [Int64]  # n_unique = 15, n_inst = 885
├─────────── b10: [List]  # n_inst = 295
│                └── [Int64]  # n_unique = 25, n_inst = 878
├── opening_id: [Int64]  # n_unique = 75, n_inst = 295
├──── openings: [List]  # n_inst = 295
│                └── [Dict]  # n_inst = 295
│                         ├─────────── w3: [List]  # n_inst = 295
│                         │                └── [Int64]  # n_unique = 15, n_inst = 701
│                         ├─────────── b2: [List]  # n_inst = 295
│                         │                └── [Int64]  # n_unique = 16, n_inst = 635
│                         ├─────────── w4: [List]  # n_inst = 295
│                         │                └── [Int64]  # n_unique = 17, n_inst = 650
│                         ├─────────── w1: [List]  # n_inst = 295
│                         │                └── [Int64]  # n_unique = 8, n_inst = 620
│                         ├──── variation: [List]  # n_inst = 295
│                         │                └── [Int64]  # n_unique = 56, n_inst = 8835
│                         ├─────────── b1: [List]  # n_inst = 295
│                         │                └── [Int64]  # n_unique = 9, n_inst = 713
│                         ├── opening_id: [Int64]  # n_unique = 75, n_inst = 295
│                         ├──────── name: [List]  # n_inst = 295
│                         │                └── [Int64]  # n_unique = 43, n_inst = 4653
│                         ├─────────── b3: [List]   # n_inst = 295
│                         │                └── [Int64]  # n_unique = 17, n_inst = 719
│                         ├─────────── b4: [List]  # n_inst = 295
│                         │                └── [Int64]  # n_unique = 19, n_inst = 619
│                         ├─────────── w2: [List]  # n_inst = 295
│                         │                └── [Int64]  # n_unique = 14, n_inst = 682
│                         └──────── code: [List]  # n_inst = 295
│                                          └── [Int64]  # n_unique = 1, n_inst = 295
├─────────── w8: [List]  # n_inst = 295
│                └── [Int64]  # n_unique = 25, n_inst = 899
├─────────── b3: [List]  # n_inst = 295
│                └── [Int64]  # n_unique = 18, n_inst = 805
├──── opening: [List]  # n_inst = 295
│                └── [Int64]  # n_unique = 44, n_inst = 4574
├────── round: [List]  # n_inst = 295
│                └── [Int64]  # n_unique = 11, n_inst = 1149
├────── black: [List]  # n_inst = 295
│                └── [Int64]  # n_unique = 55, n_inst = 4854
├─────────── w2: [List]  # n_inst = 295
│                └── [Int64]  # n_unique = 12, n_inst = 724
├─────────── w10: [List]  # n_inst = 295
│                └── [Int64]  # n_unique = 24, n_inst = 888
├─────────── b4: [List]  # n_inst = 295
│                └── [Int64]  # n_unique = 20, n_inst = 862
├─────────── b7: [List]  # n_inst = 295
│                └── [Int64]  # n_unique = 23, n_inst = 853
├──── BlackElo: [Int64]  # n_unique = 89, n_inst = 295
├──── game_id: [Int64]  # n_unique = 295, n_inst = 295
└─────────── w9: [List]  # n_inst = 295
                 └── [Int64]  # n_unique = 24, n_inst = 890
```

Figure 12: The schema of the chess dataset.

