# OpenReview forum: "Sum-Product-Set Networks: Deep Tractable Models for Tree-Structured Graphs"
_ICLR.cc/2024/Conference — ICLR 2024 poster_

### Official Review · Reviewer_9xzE · 2023-11-01

**Soundness:** 3 good
**Presentation:** 4 excellent
**Contribution:** 3 good
**Rating:** 6
**Confidence:** 4

**Summary:**

This paper proposes sum-product-set networks (SPSN), a tractable probabilistic model for tree-structured data. Interestingly, the structure of the tree and dimensionality of the data is assumed to be random rather than fixed. This is achieved through the use of random feature sets (RFS), which allows for the specification of distributions over sets of varying length taking value in some domain. RFS are integrated with Sum-Product Networks (SPN) through the use of a schema/template for the data, which hierarchically specifies the fixed parts of the tree structure (heterogenous nodes) and the variable parts (homogenous nodes). Under some (strong) assumptions on the set unit distributions and the query, inference is shown to be tractable in SPSNs. Empirical results show that the classification performance is slightly worse than, but competitive with, approaches based on neural networks.

**Strengths:**

The paper presents an (as far as I am aware) novel problem of developing a tractable probabilistic model for tree-structured data, where the graph of the tree is itself random. Such data structures naturally arise in many areas, such as XML/JSON, scientific domains, natural language (e.g. syntax trees), and relational data. The proposed solution, SPSNs, are a well-designed variant of sum-product networks that utilizes random features sets (set units) to allow tree nodes to have a random number of children while maintaining tractability.

- Novel and adept application of RFS theory to deep tractable model architectures (SPNs). This enables the specification of distributions over hierarchical random trees.
- The work could have significant impact in pushing the application of TPMs towards new domains, such as natural language processing.
- The clarity and technical quality of the paper is excellent. In particular, Figure 2 was very useful for understanding the role of sum, product and set units in relation to the tree schema.

**Weaknesses:**

- The requirement of full independence in the distribution of a set unit seems quite stringent and potentially unrealistic. For example, for the mutagenesis example in Figure 1, this would correspond to atoms in a molecule being independent (conditional on the molecule size).
- On the empirical side, to justify the importance of tractability it would be useful to test some example queries on the learned SPSNs, and their domain-specific interpretation.

**Questions:**

- Is it possible to relax the assumption of full independence in a set unit, or is this a fundamental limitation? E.g. through partial exchangability for the set unit distributions?
- Details on how SPSNs are learned seem to be missing. It seems the structure of the SPSN is fixed (Pg 4.), but how are the parameters learned? Is the cardinality distribution learned, and if so, how is it parameterized?
- Is there any technical reason why the SPSN architecture cannot be extended to DAG, rather than tree structured data?
- The related work section is thorough, but, especially considering that the tested datasets all come from a relational schema, it would be useful to better understand the relationship with relational SPNs. For example, how does one translate a relational schema to a tree schema as in Figures 4-11 (in what way is relational data "a particular form of graph-structured data")?
- It is not clear what the result is in Proposition 2 (the statement seems more like a definition).

============================
*After rebuttal/discussion*

After the author rebuttal and reviewer/AC discussion, I have mixed feelings about the work. I find the approach of SPSNs for probabilistically modelling tree structured data using RFS to be very promising, with some evidence for its utility through experiments. On the other hand, the manuscript does not fully justify/analyse key components of the approach, namely (1) providing concrete examples of interesting tractable inference queries when the number of variables is not fixed, besides marginalizing leaves (where the structure of the tree is already fixed); and (2) empirical or theoretical analysis of the impact of Assumption 1 on expressivity/modelling capacity. I still support acceptance but less enthusiastically given the mentioned weaknesses.

---

> ### Author Response · Authors · 2023-11-16
>
> *The requirement of full independence (...)*
>
> We reply on this in one of your questions below.
>
> *On the empirical side (...)*
>
> Currently, the only queries we consider are the marginal ones used for dealing with the missing values in Section 5. We will provide more queries in future work. Thank you for this constructive suggestion.
>
> *Is it possible to relax the assumption of full independence in a set unit (...)*
>
> Yes, the independence assumption can be relaxed. To satisfy our key modeling assumption, i.e., the homogeneous nodes are RFSs, the feature density would have to be represented by a sub-SPSN that is fully exchangeable under the $m!$ re-orderings of the elements of the set (and, indeed, as you suggest, the partial exchangeability can apply within the elements). This can be accomplished in two possible ways. The first one lies in modeling the interactions between the elements of the set only through the sum units, whereas the product unis will be applied consequently to model interactions withing the elements. The second one is to make the feature density fully exchangeable by marginalizing over all $m!$ permutations of the elements. This option imposes no restrictions on modeling the feature density (apart from Definition 6 to preserve the tractability), but the computational cost would be very high as discussed in Section B (Exchangeability of RFSs).
>
> Nonetheless, we do not see the independence assumption as a limitation in the context of the JSON files. The homogeneous nodes have the same schema, i.e., their structure is the same for all elements of the set; therefore, we see the elements as instances of the same random variable.
>
> *Details on how SPSNs are learned seem to be missing. It seems the structure of the SPSN is fixed (Pg 4.), but how are the parameters learned?*
>
> The details about the learning are in Section E. We perform the grid-search over the hyper-parameters that determine the structure of the SPSN network. Then, we select the best network based on the validation data.
>
> *Is the cardinality distribution learned, and if so, how is it parameterized?*
>
> Yes, the parameters of the cardinality distribution are learned through the gradient-based optimization, along with all other parameters of the network. We experimented with many variants of cardinality distributions (including mixtures), but the Poisson distribution turn out to deliver the best performance. The rate parameter of the Poisson distribution is expressed in the log-scale to allow for unconstrained optimization.
>
> *Is there any technical reason why the SPSN architecture cannot be extended to DAG (...)*
>
> We do not see any technical reason that would prevent the extension of SPSN networks to generic acyclic graphs, including directed acyclic graphs as a special case. However, such an extension will require certain modifications. In the current paper, we are motivated by the abundant use of the JSON files in practical applications (e.g., cyber-security). For this reason, we set out to focus on the tree-structured graphs. We plan to provide the extension of SPSNs to generic acyclic graphs in future work.
>
> *The related work section is thorough, but, especially considering that the tested datasets all come from a relational schema, it would be useful to better understand the relationship with relational SPNs. For example, how does one translate a relational schema to a tree schema as in Figures 4-11 (in what way is relational data ``a particular form of graph-structured data")?*
>
> This is an excellent question. We forgot to mention this in the paper. We describe this in Section G of the updated version of the manuscript.
>
> *It is not clear what the result is in Proposition 2 (...)*
>
> Thank you for noticing this, we corrected Proposition 2. Please, see the updated version of the manuscript.

---

> > ### Comment · Reviewer_9xzE · 2023-11-23
> > **Response**
> >
> > I thank the authors for their detailed response and the updated manuscript. I maintain my positive assessment of the paper.
> >
> > I appreciate the additional details about learning added to the manuscript and the clarification regarding the missing data experiments. I am still not fully convinced regarding the independence assumption not being a limitation, as it seems that exchangability rather than independence is really what you want with homogenous units - as in the mutagenesis example with molecules. But I understand that the computational cost would be high in general, and the sum units provide some expressivity in this respect.

---

### Official Review · Reviewer_snrr · 2023-11-02

**Soundness:** 4 excellent
**Presentation:** 3 good
**Contribution:** 3 good
**Rating:** 8
**Confidence:** 3

**Summary:**

The authors propose sum-product-set networks, an extension of probabilistic circuits from unstructured tensor data to tree-structured graph data. Key to their approach is the use of random finite sets to reflect a variable number of nodes and edges in the graph and allow for exact and efficient inference. The empirically demonstrate that their approach is on par with other intractable neural models.

**Strengths:**

- The paper is decently written, although, in my opinion, some important aspects seem to be missing.

- Despite the potentially toyish nature of the experiments considered, the proposed models seem to be on par with other intractable neural models, as well as significantly more robust to missing data.

**Weaknesses:**

- The writing of the paper would be greatly improved by adding informal intuitions here and there, as well as a (toy) complete example of an SPSN.

- The authors do not make it clear that one could not obtain a distribution over tree-structured graphs using e.g. knowledge compilation to compile the distribution over tree-structured graphs into a logical circuit whose parameters could then be learnt, inducing a distribution over the desired structured objects. If so, is one theoretical conclusion that there is an expressivity gap between PCs with and without set units?

- The authors only remark in passing that the infinite sum required to evaluate the set unit reduces to a finite one in practice; an argument upon which their tractability results hold. This should be made more formal.

- The writing doesn't really give us an idea of the scale of the experiments performed, but they seem toyish.

**Questions:**

- The first question that comes to mind is: can we not use knowledge compilation [1] to compile the distribution over tree-structured graphs into a logical circuit? One can use logical circuits to induce distributions over many different structured objects such as paths in a grid, hierarchies of classes, preferences, as well as subsets of size $k$ [2, 3, 4]. One can then learn the parameters of such a distribution from the data. It might very well be the case that the distribution over tree-structured graphs does not admit a tractable circuit, but such an assertion seems to be absent from the paper.

- Could you please explain what a schema is? I understand how one could obtain a schema from a tree-structured graph, but aside from the definition, I was hoping for an intuitive explanation. ( I am familiar with the term in the context of databases, which does not seem to translate? )

- Am I correct in my understanding that, according section, set units only apply to homogeneous nodes?

- I really would've like to see a (toy) complete example of an SPSN. Could you please provide such an example?

- Assumption 1 (Requirements on the set unit) "states that the cardinality distribution vanishes for a sufficiently large $m$", What exactly do you mean by that? As a follow up, am I correct in understanding that the elements of the set are independent given the cardinality? i.e. we do not consider the statistical correlations between the elements of a set? ~To me this consequently puts into questions the tractability of SPSNs laid out in proposition 1.~

- Could you please say more regarding Definition 5? What structural constraint is being imposed here exactly? To me, "follow only a single child of each sum unit" reads as determinism?

References:

[1] On probabilistic inference by weighted model counting. Mark Chavira, Adnan Darwiche. Journal on Artificial Intelligence 2008.

[2] Neuro-Symbolic Entropy Regularization. Kareem Ahmed, Eric Wang, Kai-Wei Chang, Guy Van den Broeck. UAI 2022.

[2] Semantic Probabilistic Layers for Neuro-Symbolic Learning. Kareem Ahmed, Stefano Teso, Kai-Wei Chang, Guy Van den Broeck, Antonio Vergari. NeurIPS 2022.

[3] SIMPLE: A Gradient Estimator for k-Subset Sampling. Kareem Ahmed, Zhe Zeng, Mathias Niepert, Guy Van den Broeck. ICLR 2023.

---

> ### Author Response · Authors · 2023-11-16
>
> *The writing of the paper would be greatly improved by (...)*
>
> Could you please point to specific parts of the text where these intuitions are needed? We will try to modify the manuscript in light of your suggestions. We added an example of an SPSN (Figure 4 in the updated manuscript).
>
> *The authors do not make it clear that one could not obtain a distribution over tree-structured graphs using e.g. knowledge compilation*
>
> We reply on this question below.
>
> *The authors only remark in passing that (...)*
>
> We formally specified this in Assumption 1(d) by considering $p(m)=0$ for a sufficiently large $m$. Assumption 1 is connected to our tractability result in Proposition 3. We also devoted to this a separate paragraph at the end of Section 3.1 and Remark 1 in Section B.
>
> *The writing doesn't really give us an idea of the scale of the experiments (...)*
>
> We extended Table 2 with the size of each dataset. We also comment on the computation of the experiments in the last paragraph of Section F. Moreover, we improved Figures 5-12 by including the number of instances at each level of the tree. Even though the citeseer dataset (Figure 8) has only 3312 instances (n_inst), there is approximately 300k instances (strings) in the leaf nodes.
>
> *The first question that comes to mind is (...)*
>
> We have not been aware of the existence of the approach you refer to. We find it very interesting and we would like to thank you for pointing it to our attention. We cannot see that this approach is possible in our setting, as all the references you suggest assume that the input of a PC is a fixed-size vector. Please, recall from our paper that we consider variable-size graphs. Nonetheless, it will be possible to apply such an approach if we restrict ourselves to tree-structured graphs of fixed size. We added a similar comment into the related work section.
>
> Is there an expressivity gap between PCs with and without set units? We cannot answer this question as the two models assume different inputs (the fixed size vectors versus variable-size graphs). We do not see an experiment that would allow us to compare expressivity in these two different settings.
>
> *Could you please explain what a schema is (...)*
>
> The key motivation of our paper is to create a tractable model for processing of three-structured data in the JSON format. Therefore, the definition of schema in our paper is motivated by the schema of the JSON format. It was not our intention to imply any relation to the schema used in databases. We added a footnote on this matter in Definition 2.
>
> *Am I correct in my understanding that, according section, set units only apply to homogeneous nodes?*
>
> Yes.
>
> *I really would've like to see a (toy) complete example (...)*
>
> Please, see Figure 4 in the supplementary material.
>
> *Assumption 1 (Requirements on the set unit) ``states that the cardinality distribution vanishes for a sufficiently large $m$", What exactly do you mean by that?*
>
> This means that we require $p(m)\rightarrow 0$ for $m\rightarrow\infty$. In practical cases, the homogeneous nodes contain a finite number of elements. Consequently, after learning the parameters of the cardinality distribution (the Poisson distribution in our case), we have $p(m)=0$ for $m<<\infty$, which implies that we sum over a finite number of elements in Proposition 1.
>
> For example, consider a single homogeneous node in the tree-structured graph. This node is at the same position in all instances of the training dataset (as follows from the schema). The specific value of $m$, for which $p(m)=0$, lies right after the maximum cardinality from all these instances of the homogeneous node. Alternatively, it can be restricted to a user-defined value, say, according to an allowable computational budged.
>
> *As a follow up, am I correct in understanding that the elements of the set are independent given the cardinality? (...) To me this consequently puts into questions the tractability of SPSNs (...)*
>
> Yes, as stated in Assumption 1, the elements of the set are considered independent. Could you please be more specific about why it puts the tractability of SPSNs in question?
>
> *Could you please say more regarding Definition 5? (...)*
>
> We do not assume determinism. If you convert an SPN to its mixture representation using the induced trees as in Definition 3 of [1], you can notice that each component is a product of input units with a unique scope. The union of these scopes is the scope of the root unit. The structural constrain is that $f$ must be the product of simpler functions, $\lbrace f_u\rbrace_{u\in\mathsf{I}}$, which are defined over these unique scopes. This allows the integration of $f$ to be propagated down to the input units, as the $f_u$-parts of $f$ can be integrated w.r.t. to their corresponding input units. We provide the analogous result for SPNs in Proposition 5.
>
> [1] Trapp, M., et al. 2019. Bayesian learning of sum-product networks. Advances in neural information processing systems, 32.

---

> > ### Comment · Reviewer_snrr · 2023-11-23
> >
> > Thanks you for your response. I am satisfied with most of your answers to my concerns and am updating my score accordingly.

---

### Official Review · Reviewer_k1pi · 2023-11-04

**Soundness:** 4 excellent
**Presentation:** 3 good
**Contribution:** 3 good
**Rating:** 8
**Confidence:** 3

**Summary:**

The paper suggests a new type of probabilistic circuit (PC) that can be trained and perform inference in tree-structured graph data. Standard sum-product networks (SPNs), a PC, represent a probability density over unstructured data, which forms the input random variables. To model a density over tree-structure graphs, the manuscript introduces "set units," nodes in the PC that allow for a variable number of nodes/edges in the data graph.

**Strengths:**

* Tractability and Exchangeability: The manuscript presents a theoretical foundation for the tractability of the method, based on PC results in Section 3.1 and SPSNs' exchangeability based on its node types in Section 3.2.
* Using the theory of finite random sets in SPSNs yields a simple and elegant way of representing tree-structured graphs.

**Weaknesses:**

* Implementation: the manuscript does not provide a transparent discussion about the implementation of SPSNs. The non-formal description provided in "Building SPNs" raises relevant questions related to the convergence and size of the model. Moreover, from the setup, it is unclear how sensitive parameter initialization and/or hyper-parameter tuning the model is.
* Experiments are encouraging but not convincing. Section 5 is unclear on how the tractability and exchangeability properties of SPSNs are exploited in the experiments. While the missing values results in Figure 3 are beneficial, they do not highlight "efficient inference over
specific parts of the data graph," as stated in the Conclusion. Moreover, it might be helpful to the manuscript to compare the results with recent works, such as the ones discussed in Section 4.
* Paper presentation
    - The writing is unclear between PCs and SPNs, as the title evokes SPNs while the text uses PCs. The authors should clarify the difference between the two or assume interchangeable usage under assumptions.
    - It could be beneficial to discuss the differences between SPNs and SPSn sooner in the paper, as it is a key contribution of the work. The sentence "This differs from the conventional sum-product network..." is helpful but only appears in Section 3.
    - The manuscript could better articulate its motivation by connecting the problems presented in the introduction with some of the results. For instance, it is not clear in the paper how SPSNs take advantage of "the parent-child ancestry inherent in tree-structured graphs" in a different way than competitive generative models.

**Questions:**

* Were there any empirical boundaries or assumptions when implementing the recursive algorithm described in "Building SPSNs"? How do you deal with large data graphs with multiple heterogeneous nodes and size constraints?
* Could you please expand on the "(...) building a set of trees based on a user-specified neighborhood" in Section 4 regarding the similar graph-based approach from (Errica & Niepert, 2023)?

---

> ### Author Response · Authors · 2023-11-16
>
> *Implementation: the manuscript does not provide a transparent discussion about the implementation (...)*
>
> We added Section E into the updated manuscript. It contains a more detailed description of constructing SPSNs. We also included comments on the size of each SPSN block in the network.
>
> *Section 5 is unclear on how the tractability and exchangeability properties of SPSNs are exploited in the experiments.*
>
> We added comments into the last paragraph of Section 5 that will better connect our claims about tractability with the experiments.
>
> *While the missing values results in Figure 3 are beneficial, they do not highlight ``efficient inference over specific parts of the data graph," as stated in the Conclusion.*
>
> We decided to demonstrate the tractable inference of SPSNs on the task of marginalizing the missing values in the leaf nodes. This corresponds to marginalizing out the whole leaf nodes, which we see as a part of the data graph. Our motivation to show the tractability on the marginalization task lies in that it is at the core of many more advanced queries, implying that they can also be evaluated tractably.
>
> *Moreover, it might be helpful to the manuscript to compare the results with recent works.*
>
> The related work on the specific type of tree-structured graphs considered in our case is rather sparse to the best of our knowledge. We assume the tree-structured graphs where the inner nodes have no feature vectors and the leaf nodes have feature vectors with varying dimensionality and data type. We found the most related approaches to this setting in the NLP domain. They are mentioned in Section 4 and compared to our approach in Section 5. The remaining methods discussed in Section 4 assume generic (undirected and cyclic) graphs with all nodes containing feature vectors of the same dimension. Though they are principally applicable to tree-structured graphs, there would be the need to make non-trivial adaptations to deal with the empty features and varying dimensions.
>
> *The writing is unclear between PCs and SPNs (...)*
>
> Thank you for pointing this to our attention. We revised the usage of PCs throughout the text. We consider SPNs as a sub-class of the large family of PCs. We included a comment on this matter into third paragraph of Section 1. Please, note that sometimes we use the term PCs deliberately since we need to refer to certain properties of SPSNs that are inherited from PCs.
>
> *It could be beneficial to discuss the differences between SPNs and SPSn sooner in the paper (...)*
>
> We added a comment on this matter into the third paragraph of Section 1.
>
> *The manuscript could better articulate its motivation by connecting the problems presented in the introduction with some of the results. For instance, it is not clear in the paper how SPSNs take advantage (...)*
>
> We elaborated on this in the second and fourth paragraph of Section 1.
>
> *Were there any empirical boundaries or assumptions when implementing the recursive algorithm described in ``Building SPSNs"? How do you deal with large data graphs with multiple heterogeneous nodes and size constraints?*
>
> Line 6 in the procedure scope_layers of Algorithm 2 repeatedly checks if the set of scopes $\psi$ contains a singleton set or if we reached a prescribed number of layers of the SPSN block. If any of these two conditions holds, then the procedure stops to create more layers in the SPSN block. This means that if $n_l$ and the heterogeneous nodes are large, the SPSN block can grow exponentially in size (similarly to the conventional SPN architectures). We provide concrete formulae to compute the number of parameterized computational units of the SPSN block in the last paragraph of Section E. In our experiments, we set the boundary on the computational budged by fixing $n_l$ to a given value.
>
> If we set a reasonably small $n_l$ to limit the computational budget and the heterogeneous nodes are still large after performing several splits in the product layers, we apply the full independence assumption to model the elements of the heterogeneous nodes in the input units (which we do not present in Algorithms 1 for simplicity). This is a common practice when applying conventional SPNs to high-dimensional vector data, especially if we use the binary split in the product units [1].
>
> [1] Peharz, R., et al. 2020. Random sum-product networks: A simple and effective approach to probabilistic deep learning. In Uncertainty in Artificial Intelligence (pp. 334-344). PMLR.
>
> *Could you please expand on (...) Errica \& Niepert, 2023?*
>
> The authors decompose generic cyclic graphs into a collection of trees. Then, they use each tree as a template to design a hierarchy of SPNs. In this hierarchy, there is a single SPN per each node in the tree in order to model the feature vector of that node. The tree has a height $L$, which is a user-defined hyper-parameter defining the neighborhood of a selected vertex $v$ in the original graph.

---

> ### Author Response · Authors · 2023-11-23
>
> Dear Reviewer k1pi,
>
> Your valuable recommendations have allowed us to make several enhancements to the paper. Once again, thank you for investing your time and experience in helping us improve our work. As the discussion period ends soon, we kindly ask if you have considered increasing the rating based on our responses.
>
> Kind regards,
>
> Authors

---

### Official Review · Reviewer_gTbZ · 2023-11-08

**Soundness:** 3 good
**Presentation:** 3 good
**Contribution:** 3 good
**Rating:** 6
**Confidence:** 2

**Summary:**

This paper develops a sum-product-set networks (SPSNs) to study the tree-structured graphs. It develops new variant of probabilistic circuts to obtain tractable inference for SPSNs. In the experiments, it shows that SPSNs obtains comparable performance to Neural Networks in the graph classification task.

**Strengths:**

The paper has the following strengths:

**1** the problem this paper working on is important to the community. The way of extending the applicability of probabilistic circut to the tree-structured graph data would be interesting and promising to the community.

**2** the presentation and writing are very well. Although I am new to this topic, I can easily understand the main points and the main mechanism of this SPSNs method.

**3** I like the investigation of the exchangeability of SPSNs. The study seems complete.

**Weaknesses:**

Due to my limited expertise, I did not have identified meaningful weaknesses.

**Questions:**

Sorry I am not an expert in this topic, I did not have particular technical questions.

---

> ### Author Response · Authors · 2023-11-16
>
> We would like to thank you for seeing our paper as easily understandable even for newcomers to the field of probabilistic circuits.

---

### Author Response · Authors · 2023-11-21

Dear Reviewers,

We are very grateful for all your constructive and thorough comments on our paper. We have considered your concerns and suggestions and improved the paper as described below. We will gladly answer any further questions.

As the discussion period is coming to a close, please consider our responses and potentially increase the chance of accepting the paper by raising its rating.

Thank you again for all your time and effort in helping us improve the paper. We look forward to hearing from you soon.

Kind regards,

Authors

---

### Meta-Review · Area_Chair_5We3 · 2023-12-10

**Metareview:**

This paper tackles the problem of learning a tractable probabilistic model for tree-structured data. Specifically, authors are proposing an extension of probabilistic circuits that, by introducing a set unit in their graphs, is able to represent intermediate exchangeable distributions. The proposed architecture is trained on standard graph benchmarks and is on par with simple (intractable) neural baselines on tasks such as graph classification.

Reviewers appreciated the direction of this contribution, and especially the formalization of the set units. At the same time, in their reviews they highlighted how presentation might be missing details on how to build and learn the circuits, as well as discussions w.r.t. the limitations of tractable set distributions and the relationships with previous exchangeable models. The authors provided more details in the rebuttal but the questions about the limitations and comparisons (e.g. w.r.t. RSPNs) are still open. That being said, the paper still constitutes a nice contribution to the landscape of tractable probabilistic models.

The paper is accepted and authors are asked to comment in detail in the camera ready on the limitations of the tractability for set distributions and highlight the relationship with RSPNs beyond saying they do not support distributions over cardinality, and eliciting the different treatment of exchangeability.

**Justification For Why Not Higher Score:**

While the proposed TPM is an interesting addition, the use for tractable inference on graphs is not properly justfied in the paper or in the experiments.

**Justification For Why Not Lower Score:**

The contribution is still valuable as a first work on tractable distributions over tree structured data.

---

### Decision · Program_Chairs · 2024-01-16

Accept (poster)